# Historical variation in normalized difference vegetation index compared with soil moisture at a taiga forest ecosystem in northeastern Siberia

Aleksandr Nogovitcyn[1], Ruslan Shakhmatov[1,2], Tomoki Morozumi[3], Shunsuke Tei[4,5], Yumiko Miyamoto[4,6], Nagai Shin[7], Trofim C. Maximov[8] and Atsuko Sugimoto[4]

[1]Graduate School of Environmental Science, Hokkaido University, Sapporo, 060-0817, Japan

[2]Slavic-Eurasian Research Center, Hokkaido University, Sapporo, 060-0809, Japan

[3]National Institute for Environmental Studies, Tsukuba, 305-8506, Japan

[4]Arctic Research Center, Hokkaido University, Sapporo, 001-0021, Japan

[5]Forestry and Forest Products Research Institute, Tsukuba, 305-8687, Japan

[6]Faculty of Agriculture, Shinshu University, Kamiina, 399-4598, Japan

[7]Research Institute for Global Change, Japan Agency for Marine-Earth Science and Technology, Yokohama, 236-0001, Japan

[8]Institute for Biological Problems of Cryolithozone, Siberian Branch of the Russian Academy of Sciences, Yakutsk, 677000, Russia

*Correspondence to*: Atsuko Sugimoto (sugimoto@star.dti2.ne.jp)

**Abstract.** The taiga ecosystem in northeastern Siberia, a nitrogen-limited ecosystem on permafrost with a dry climate, changed during the extreme wet event in 2007. We investigated the normalized difference vegetation index (NDVI) as a satellite-derived proxy of needle production and compared it with ecosystem parameters such as soil moisture water equivalent (SWE), larch foliar C/N ratio, $\delta^{13}C$ and $\delta^{15}N$, and ring width index (RWI) at the Spasskaya Pad Experimental Forest Station in Russia for the period from 1999 to 2019. Historical variations in NDVI showed a large difference between typical larch forest (unaffected) and the sites affected by the extreme wet event in 2007 because of high tree mortality at affected sites under extremely high SWE and waterlogging, resulting in a decrease in NDVI, although there was no difference in the NDVI between typical larch forest and affected sites before the wet event. Before 2007, the NDVI in a typical larch forest showed a positive correlation with SWE and a negative correlation with foliar C/N. These results indicate that not only the water availability (high SWE) in the previous summer and current June but also the soil N availability likely increased needle production. NDVI was also positively correlated with RWI, resulting from similar factors controlling them. However, after the wet event, NDVI was negatively correlated with SWE, while NDVI showed a negative correlation with foliar C/N. These results indicate that after the wet event, high soil moisture availability decreased needle production, which may have resulted from lower N availability. Foliar $\delta^{15}N$ was positively correlated with NDVI before 2007, but after the wet event, foliar $\delta^{15}N$ decreased. This result suggests damage to roots and/or changes in soil N dynamics due to extremely high soil moisture. As a dry forest ecosystem, taiga in northeastern Siberia is affected not only by temperature-induced drought but also by high soil moisture, led by extreme wet events, and nitrogen dynamics.

## 1 Introduction

Boreal forests in northern regions of North America and Eurasia, including islands, occupy a large area, approximately 27 % of the world's forest cover (FAO, 2020). Under conditions of increasing atmospheric $CO_2$ concentrations (e.g. Friedlingstein et al., 2022), the role of taiga and other terrestrial ecosystems as carbon sinks becomes more important. Among the taiga areas, Alaska, Canada, and Siberia are distinguished by permafrost, which is one of the main components of the global carbon cycle. Under warming conditions, northern ecosystems respond differently to environmental changes depending on the different biomes and regions, as mentioned below from remote sensing satellite observations. The normalized difference vegetation index (NDVI), the most common remote sensing vegetation data, typically shows increasing trends (greening) in arctic tundra but decreasing trends (browning) in many boreal forests (Bunn and Goetz, 2006; Verbyla, 2008; Berner and Goetz, 2022; Miles and Esau, 2016). Observed greening at the tundra and taiga-tundra boundary can be associated with the expansion of trees and shrubs to the north (Frost and Epstein, 2014; Tape et al., 2006; Shevtsova et al., 2020) and increases in aboveground plant biomass (Goetz et al., 2005; Berner et al., 2013; Forbes et al., 2010). In turn, the browning of taiga can be attributed to tree mortality and decreases in forest production because of droughts (Welp et al., 2007; Bunn and Goetz, 2006), forest fires (Goetz et al., 2005), and extreme wet events (Famiglietti et al., 2021; Nogovitcyn et al., 2022). Nevertheless, temporal changes in the NDVI of boreal forests are spatially heterogeneous and vary depending on different factors, such as the plant vegetation types (Myers-Smith et al., 2020; Bunn and Goetz, 2006), stand density (Bunn et al., 2007; Dearborn and Baltzer, 2021) and topography (Sato and Kobayashi, 2018). One of the most important factors controlling greenness is water availability (Ruiz-Perez and Vico, 2020; Forkel et al., 2015). Overall, warming condition of boreal forests has positive effects in wetter regions but negative effects on forest productivity in dry regions (e.g., Ruiz-Perez and Vico, 2020).

Siberian taiga is mainly covered with deciduous conifers, larches, which grow under severe conditions, such as continental climate, that is, cold winters, hot summers, low precipitation (Archibold, 1995), and limited nitrogen availability (Popova et al., 2013; Kajimoto et al., 1999). Permafrost and seasonal ice are important sources of water for larches during drought (Sugimoto et al., 2003; Sugimoto et al., 2002). These severe conditions make this ecosystem vulnerable to environmental changes. In northeastern Siberia, the role of the taiga ecosystem and its responses to climate change have been studied at the Spasskaya Pad Forest Station near Yakutsk over a long-term. Over the past few decades, the ring width index (RWI) has decreased in this region as well as in other continental dry regions, Alaska, Canada, and southern Europe, because of high temperature-induced droughts (Tei et al., 2017). On the other hand, precipitation extremes, which are predicted to be more intensive and frequent (Douville et al., 2021; Wang et al., 2021), can also negatively affect the forest. In addition to droughts, extreme wet events occur because of intensive precipitation, such as heavy rainfall and snowfall. In this forest area, changes in winter precipitation may shift the forest phenology and production of soil inorganic nitrogen, as demonstrated in snow manipulation experiments (Shakhmatov et al., 2022).

In northeastern Siberia, boreal forests have been affected by drought in the past because of the continental climate. However, in 2007, soil moisture was the highest in the past century (Tei et al., 2013) because of large amounts of rainfall and subsequent winter snowfall, which led to waterlogging in topographic depressions of the forest. This extreme wet event was fatal for many trees in the forest, especially in the depressions (Iijima et al., 2014; Ohta et al., 2014). These extremely moist conditions reduced $CO_2$ uptake by vegetation (gross primary production) and evapotranspiration (Ohta et al., 2014). Evapotranspiration in this region is mainly controlled by the leaf area index (LAI) (Matsumoto et al., 2008), suggesting a decrease in LAI in 2007 because of high tree mortality. However, after the extreme wet event, eddy-covariance flux measurements showed no significant trends in $CO_2$ exchange at the ecosystem scale (Kotani et al., 2019). The affected sites are distinguished not only by high tree mortality but also by a secondary succession of the understory and floor vegetation communities to water-resistant species, which may have compensated for the reduced water and $CO_2$ fluxes (Ohta et al., 2014).

Boreal forests in northeastern Siberia have been affected by both drought and extreme wet events, as described above, which involved complex changes in the forest environment. Tei et al. (2019a) demonstrated that larch trees, which died during the extreme wet event, were affected by a previous drought. However, at the forest level, the NDVI was not related to gross primary production in 2004–2014 (Tei et al., 2019b), and the NDVI of the larch forest damaged by waterlogging showed insignificant trends in 2000–2019 and an insignificant correlation with climate data (Nagano et al., 2022). These phenomena were attributed to waterlogging-induced changes in the understory and floor vegetation composition (Nagano et al., 2022; Tei et al., 2019b), but they seem to be also related to changes in the conditions of the larch forest.

The taiga ecosystem in eastern Siberia is one of the most important biomes in the world and is vulnerable to climate change. The purpose of this study was to understand how the larch forest in this region changed, particularly after the extreme wet event in 2007, and what factors impacted the changes over the past two decades. Therefore, we investigated historical variations in satellite-derived NDVI to evaluate changes in the forest, as the leaves of deciduous trees reflect the condition of trees in each year. To understand the factors impacting forest changes, especially related to the extreme wet event, NDVI data were compared with field-observed parameters in a typical larch forest, such as the RWI, soil moisture, needle $\delta^{13}$C, $\delta^{15}$N, C/N, air temperature, and precipitation from 1998 to 2019.

## 2 Materials and Methods

### 2.1 Study site

The study was conducted in the Spasskaya Pad Experimental Forest (62°15'18''N, 129°37'08''E, alt. 220 m a.s.l.), Institute of Biological Problems of Cryolithozone, Siberian Branch of the Russian Academy of Sciences (IBPC SB RAS), near Yakutsk, Russia (Fig. 1a). The region in Eastern Siberia is established on continuous permafrost and has a continental climate

(dry climate) with an extremely high annual temperature range. During the observation period from 1991 to 2020 at Yakutsk,
the average annual precipitation was 233 mm, and the average monthly temperature ranged from -37˚C to +20 °C in cold
January and warm July, respectively. Dominant species is larch (*Larix cajanderi*) that is deciduous conifer (Abaimov et al.,
1998), mixed with broadleaved birch (*Betula pendula*), and the understory includes small shrubs, such as evergreen cowberry
(*Vaccinium vitis-idaea*) and deciduous bearberry (*Arctous alpina*), and grasses.
During the period from 2005 to 2007 (water years, from October to September), there was a large amount of precipitation
(307 ± 29 mm) continuously, which caused a significant increase in soil moisture (Sugimoto, 2019) and even waterlogging.
Consequently, an extreme wet event occurred in 2007, which damaged larch forests, resulting in high tree mortality and a
change in the composition of understory vegetation to moisture-tolerant grasses and shrubs in some areas, especially in
depressions (Iijima et al., 2014; Iwasaki et al., 2010; Ohta et al., 2014). In the summer of 2018, we set a 60 m × 510 m
transect, which included areas unaffected and affected by the extreme wet event (Fig. 1a, b). The transect was divided into 30
× 30 m plots (34 plots in total). Using these plots, we observed spatial variation in NDVI (Nogovitcyn et al., 2022). In this
study, we visually classified the forest conditions based on photographs. Four forest types were identified along the transect
(Fig. 1c): typical mature (TF; number of plots in the transect, $n = 17$), regenerating-1 (RF-1; $n = 11$), regenerating-2 (RF-2; $n$
$= 4$), and damaged (DF; $n = 2$) forests. The first TF showed no visible damage from the extreme wet event. The plots
discerned as regenerating forests RF-1, had many dead mature larches and formed forest gaps in the overstory where there
were a large number of young larches (seedlings and saplings with a height of up to 3 m) and shrubs. Regenerating forests,
RF-2, contained more dead mature larches and more young larches compared to those in RF-1. Damaged forests, DF, where
all mature trees died, was predominantly covered by moisture-tolerant grasses, and had much smaller numbers of young
larches than in RF-1 and RF-2. The DF plots were located on a depression in a trough-and-mound topography, and some
patches of the DF plots were flooded.

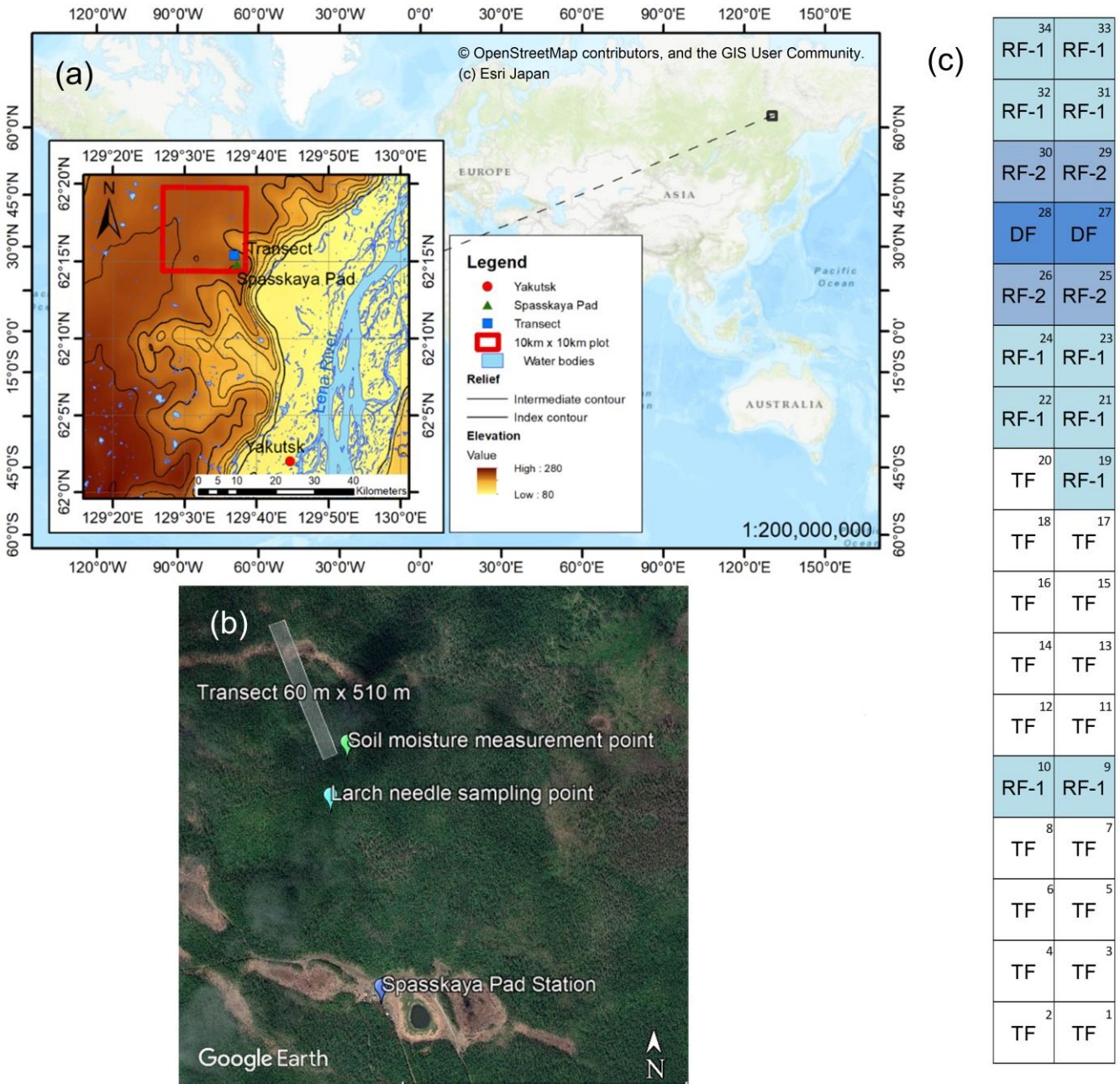

**Figure 1.** (a) Location of the Spasskaya Pad Station (62°15'18"N, 129°37'08"E) and the study transect near Yakutsk in the topographic map (modified from USGS/NASA Landsat 8 image) zoomed from a global map (Source: © OpenStreetMap contributors, © Esri Japan). (b) Detailed view of the study area in the Spasskaya Pad Forest (Source: © Google Earth, © 2022 Maxar Technologies): locations of the station, 60 m x 510 m transect, and points of soil moisture measurement and larch needle sampling for $\delta^{13}C$, $\delta^{15}N$, and C/N. (c) Scheme of the transect with a total of 34 plots, which were divided into four forest types based on the level of forest damage (Nogovitcyn et al., 2022): typical forests (TF), two types of regenerating forests (RF-1 and RF-2), and damaged forests (DF).

## 2.2 NDVI

The raster normalized difference vegetation index (NDVI) was computed based on the Landsat Collection-1 Level-2 image products (https://earthexplorer.usgs.gov/) with a spatial resolution of 30 m using QGIS software (v. 3.2.2-Bonn):

$$NDVI = (NIR - R) / (NIR + R),$$

where NIR and R are the near-infrared and red surface reflectance bands of the product, respectively. The image products georeferenced to the WGS-84 UTM 52N coordinate system were selected according to the location of the study transect. The NDVI value was extracted for each transect plot using the zonal statistics function. The transect plots, which consist of only pixels attributed to quality pixels (clear terrain, low-confidence cloud, and low-confidence cirrus) in the quality assessment bit index band according to Landsat Surface Reflectance product guides, were used in the analysis.

To investigate the historical variation in NDVI, we considered the seasonal maximum of the mean NDVI of the transect for the long-time period from 1999 to 2019. The longest time-series data available for the study area has been obtained by the Landsat 7 satellite with the Enhanced Thematic Mapper Plus (ETM+) image sensor since 1999. However, its sparse temporal resolution (16 days) and scan-line corrector failure in 2003 forced the consideration of additional data from other satellites, such as Landsat 5 Thematic Mapper (TM) (available until 2011) and Landsat 8 Operational Land Imager (OLI) (available since 2013). Because the last two have different sensors in contrast to Landsat 7, NDVI values calculated from the TM and OLI images were converted to ETM+ using the linear equations:

$$NDVI_{ETM+} = 1.037 \cdot NDVI_{TM},$$

$$NDVI_{ETM+} = 0.9589 \cdot NDVI_{OLI} + 0.0029$$

developed by Ju and Masek (2016) and Roy et al. (2016), respectively, for boreal forests. Local parametrization of signals from different sensors was not performed for our study site because of the insufficient overlap in the acquired images. We identified that the selected and converted data were close to the 1:1 lines between Landsat 5 and 7 and between Landsat 7 and 8. For each year, a paired sample *t*-test was applied to determine the difference between the mean NDVI of the transect on the observation days. In the case of statistically insignificant differences among observation days, we selected the day with the highest number of quality pixels.

To verify the historical variation in NDVI of the transect, a larger area, 10 km × 10 km (hereafter, the 10-km plot), including the Spasskaya Pad Forest, was used for comparison with the transect (the center of the 10-km plot was located at 62°17'4"N, 129°32'44"E; Fig. 1a). For each observation day, the mean NDVI of the 10-km plot was calculated using only quality pixels using ENVI 5.1 (L3Harris Technologies, USA). For each year, the seasonal maximum NDVI of the 10-km plot was determined as the highest mean NDVI among observation days, on which the number of quality pixels were more than 50 % (total, 111,556 pixels). The seasonal maximums of the transect and 10-km plot showed the same day for about three-quarters of the study period (15 years among 21) and showed a different day in six years (Table S1): 2006 (7 August and 29 July), 2007 (1 and 25 July), 2010 (1 and 15 July), 2011 (5 and 12 August), 2015 (23 and 31 July), and 2019 (1 and 9 July). The averaged NDVI values of the 10-km plot, transect, and each forest type (TF, RF-1, RF-2, and DF) in the transect are shown in Fig. 2a and 2b.

The NDVI data of larch forest, which is deciduous, quickly increases in early summer, when the NDVI is mostly stable (e.g.,
Huete et al., 2002). This stable NDVI continues for more than 1.5 months (typically from July to mid-August), although the
time period depends on the weather and soil moisture conditions. Seasonal maximum NDVI was identified during this
period. Although the data acquisition days were limited because of the low temporal resolution and cloud coverage, more
than three days of data were acquired by combining three satellite images, and seasonal maximums were determined, except
for in 1999 and 2003. These two years had only one data acquisition day, on 27 August 1999 and 21 July 2003, and both data
points were recognized as the seasonal maximum.

## 2.3 Ecosystem and climate parameters

Several ecosystem parameters have been observed since 1998 in typical forests. To monitor the physiological response of
larch to environmental changes, the C and N isotopic compositions ($\delta^{13}C$ and $\delta^{15}N$, ‰), and the ratio of C to N content (C/N)
of larch needles have been observed since 1999, except in 2012, at the site 200 m south of the transect (Fig. 1b). The $\delta^{13}C$
and $\delta^{15}N$ are calculated by:

$$\delta^{13}C \text{ (or } \delta^{15}N) = (R_{sample}/R_{std} - 1) \times 1000 \text{ (‰)},$$

where $R_{sample}$ and $R_{std}$ are isotope ratios ($^{13}C/^{12}C$ or $^{15}N/^{14}N$) of the sample and standard, respectively, and standards are
Vienna Peedee Belemnite for C and atmospheric $N_2$ for N. The foliar $\delta^{13}C$ reflects the physiological condition of
photosynthesis and is widely applied to indicate plant water use efficiency (Farquhar et al., 1989). The $\delta^{13}C$ value of plant
tissue (e.g., leaf) is expressed using the following equation:

$$\delta^{13}C = \delta^{13}C_{atm} - a - (b - a) (C_i/C_a),$$

where $\delta^{13}C_{atm}$ is the C isotopic composition of atmospheric $CO_2$, a (4.4‰) and b (27‰) are isotope fractionations of the
diffusion and photosynthetic reaction, and $C_i$ and $C_a$ are intercellular and atmospheric $CO_2$ concentrations. The foliar $\delta^{13}C$
becomes high when higher irradiance and lower stomatal conductance are observed. At our study site, lower and higher $\delta^{13}C$
values of larch needles were typically due to wet and drought conditions. The foliar $\delta^{15}N$ is a physiological indicator of the N
source for a plant (Evans, 2001), which can vary depending on numerous physiological and environmental factors. The foliar
C/N represents the N status of a plant (Liu et al., 2005).
Larch needles were collected from four to eight young larch trees in August every year. We collected them from the same
trees (sample trees) every August, and these trees are located nearby to each other. Four stems were obtained from each tree,
and the needles from each tree were mixed and analyzed at Kyoto University (samples for 1999–2003) and Hokkaido
University (samples after 2004) using Conflo systems (EA 1108 and Delta S, and Flash EA 1112 and Delta V, Thermo Fisher
Scientific, at Kyoto and Hokkaido Universities, respectively). Analytical precisions (standard deviation) of the C and N
content measurements were better than 0.3% and 0.1%, respectively, and those for the isotopic compositions $\delta^{13}C$ and $\delta^{15}N$
were better than 0.2‰. The details of sampling, sample preparation and laboratory analyses of C and N contents and their
isotope compositions using the EA-IRMS system are described in Fig. S1. The details of the average calculations are shown
in Fig. S1 and S2. In 2015, there were no data on the $\delta^{15}N$ and N content.
For more than 100 years, until 2016, larch ring-width index (RWI) indicating wood growth dynamics was estimated by
detrending and standardizing the raw time-series width data obtained from the collected paired cores (Tei et al., 2019b; Fan et
al., 2021). The RWI data used for analysis are shown in Table S2.
Soil moisture was measured using time-domain reflectometry (TDR), and the soil moisture water equivalent (SWE; the
amount of liquid water contained within the soil layer, mm) in the 0–60 cm soil layer was obtained for 1998–2019 in June,
July, and August using the method described by Sugimoto et al. (2003) (near the transect; Fig. 1b). There were no data for
June of 2002 and 2011 and August of 2003. The details of the intra- and inter-annual variations in SWE are shown in Fig. S3.
Among the climate variables, summer air temperature and precipitation datasets recorded by the meteorological station at
Yakutsk (62.02° N, 129.72° E) were obtained from the All-Russia Research Institute of Hydrometeorological Information -
World Data Centre (RIHMI-WDC) website (http://aisori-m.meteo.ru/).
**2.4 Statistical analysis**
All statistical analyses were carried out using R statistics v.4.1.3 (R Core Team). Relationships between datasets were
investigated using a simple linear regression model (function "lm") and a Pearson correlation test ("cor.test"), the most
common statistical test based on the method of covariance. Trends of NDVI change in 1999–2019 were estimated using the
Mann–Kendall test (package "trend", function "mk.test"). Differences in NDVI among four forest types (TF, RF-1, RF-2 and
DF) were determined using Kruskal-Wallis test ("kruskal.test") with pairwise Wilcoxon rank sum test
("pairwise.wilcox.test"). The results of the statistical tests are shown in the Supplemental (Table S3–S9). The models and
tests described by levels of statistical significance ($p$-values) less than 0.05 and 0.1 were considered to be "significant" and
"moderately significant", respectively.
**3 Results**
**3.1 Year-to-year variation of seasonal maximum NDVI**
Fig. 2a shows the historical variation in the seasonal maximum NDVI of the TF and the 10-km plot from 1999 to 2019. Both
NDVI time series varied similarly. The seasonal maximum of each year was observed from 25 June to 13 August, except for
1999 (shown in Table S1). The maximum transect NDVI in 1999 was observed on 27 August ($0.75 \pm 0.02$, $n = 34$) because
the Landsat data in 1999 were limited to the latter half of August. The mean seasonal maximum NDVI for the transect varied
between 0.72 and 0.80. During the period from 1999 to 2001, the NDVI of the transect was high from $0.75 \pm 0.02$ ($n = 34$) to
$0.80 \pm 0.02$ ($n = 34$) (Fig. 2a), but in 2002 and 2003, the NDVI was much lower ($0.73 \pm 0.02$, $n = 34$) than that in 2001. From

216 2003 to 2006, NDVI again increased from $0.73 \pm 0.02$ ($n = 34$) to $0.76 \pm 0.02$ ($n = 34$). During the wet event in 2007–2008,

217 the NDVI decreased to $0.73 \pm 0.04$ ($n = 34$). After 2009, NDVI was higher than that in 2008 ($0.72 \pm 0.03$, $n = 34$), except in

218 2016 ($0.72 \pm 0.03$, $n = 31$).


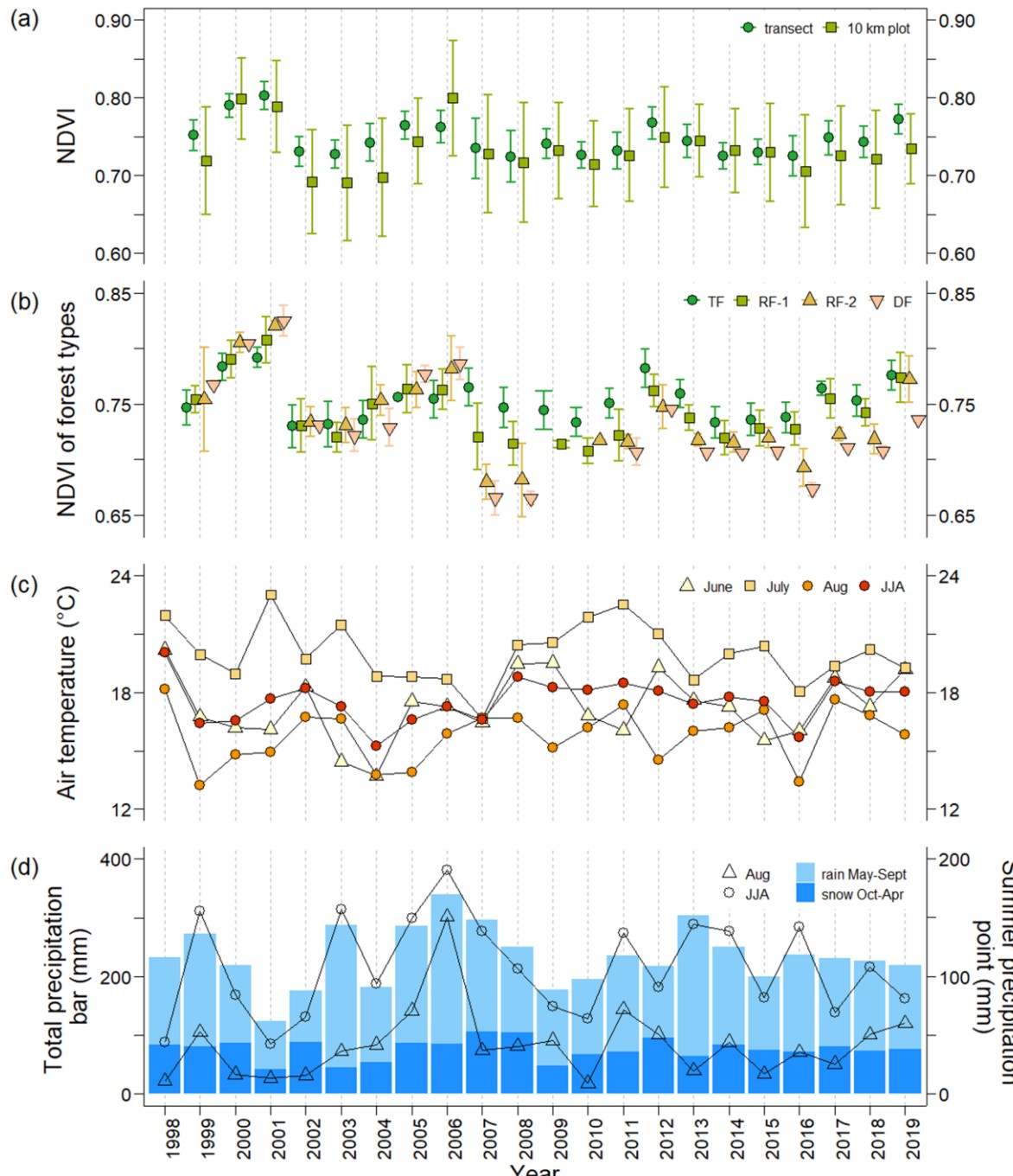


**Figure 2.** The temporal variations from 1999 to 2019 in (a) seasonal maximum NDVI averaged for the plots in the transect and the
representative 10-km forest plot calculated from available Landsat 5, 7, 8 images; (b) NDVI of four forest types, typical mature forest (TF),
regenerating forests (RF-1 and FR-2), and damaged forest (DF); (c) mean air temperature in June, July, August and whole summer period
JJA (June-July-August); (d) the amount of precipitation during previous October-current April (snow) and current May-September (rain)
shown with blue bars, in August and whole summer period JJA (June-July-August) shown with triangles and circles. Both the mean NDVI
of the 10-km plot and the transect decreased from 1999 to 2019 (-0.0009 and -0.0010 year$^{-1}$, respectively), but not with statistically
significant trends. Generally, the mean NDVI in the transect was higher than that in the 10-km plot, except for 2000 and 2014.

**3.2 NDVI of each forest type**

The NDVI time series for four forest types (typical forest TF, regenerating forests RF-1 and RF-2, and damaged forest DF) in the transect during 1999–2019 are shown in Fig. 2b. As shown in Fig. 2b and 3a, before 2007, the NDVI of TF during 1999–2001 ($0.75 \pm 0.02$ to $0.79 \pm 0.01$, $n = 17$) was higher than that in the subsequent period, 2002–2006 ($0.73 \pm 0.02$ to $0.75 \pm 0.02$, $n = 17$). In 2002, there was a significant decrease in the TF NDVI, which remained low between 2002 and 2004 (Fig. 2b and 3a). During 1999–2006, the NDVI values of the four types were close to each other, but after the wet event, NDVI values noticeably differed among the forest types (Fig. 2b). In 2007, the NDVI of TF ($0.76 \pm 0.02$, $n = 17$) was the highest, and those of the other three types decreased in the order of RF-1 ($0.72 \pm 0.03$, $n = 11$), RF-2 ($0.68 \pm 0.02$, $n = 4$), and DF ($0.67 \pm 0.02$, $n = 2$) (Fig. 2b). In 2008, the NDVI decreased slightly and showed the same order of forest types as that in 2007. After 2009, the difference among the forest types, especially between TF and DF, remained, although it was smaller than that in 2007.

**3.3 NDVI of the typical forest and ecosystem parameters of the study site**

To consider the historical variation in the NDVI of typical forests in our study area, the TF NDVI and observed parameters were compared (Fig. 2 and 3). In Fig. 4, the linear relationships between NDVI and other parameters were investigated for two different time periods, before (1999–2006) and after (2008–2019), to compare them.

**3.3.1 Climate parameters (temperature and precipitation) at Yakutsk**

Interannual variations in climatic parameters, such as air temperature and precipitation, from 1999 to 2019 are displayed in Fig. 2c and 2d. The average air temperature in June–August (summer temperature) was relatively high in 1998, 2001–2002, and 2008–2012 (Fig. 2c). The TF NDVI showed no correlation with summer temperature. The amount of annual water precipitation (from October to September) for the period from 1991 to 2020 averaged approximately $233 \pm 47$ mm. As shown in Fig. 2d, larger water year precipitation, that is, precipitation higher than 280 mm (one standard deviation above the mean for 1991–2020), was observed in 2003 (287 mm), 2005–2007 (285, 340, and 296 mm), and 2013 (304 mm). The amount of water precipitation in 2001 (124 mm) was the lowest during the observation period. The drought year (2001) showed a high TF NDVI value. Consecutive wet years in 2005–2007 showed slightly higher TF NDVI values, but there was no correlation between the water year precipitation and NDVI.

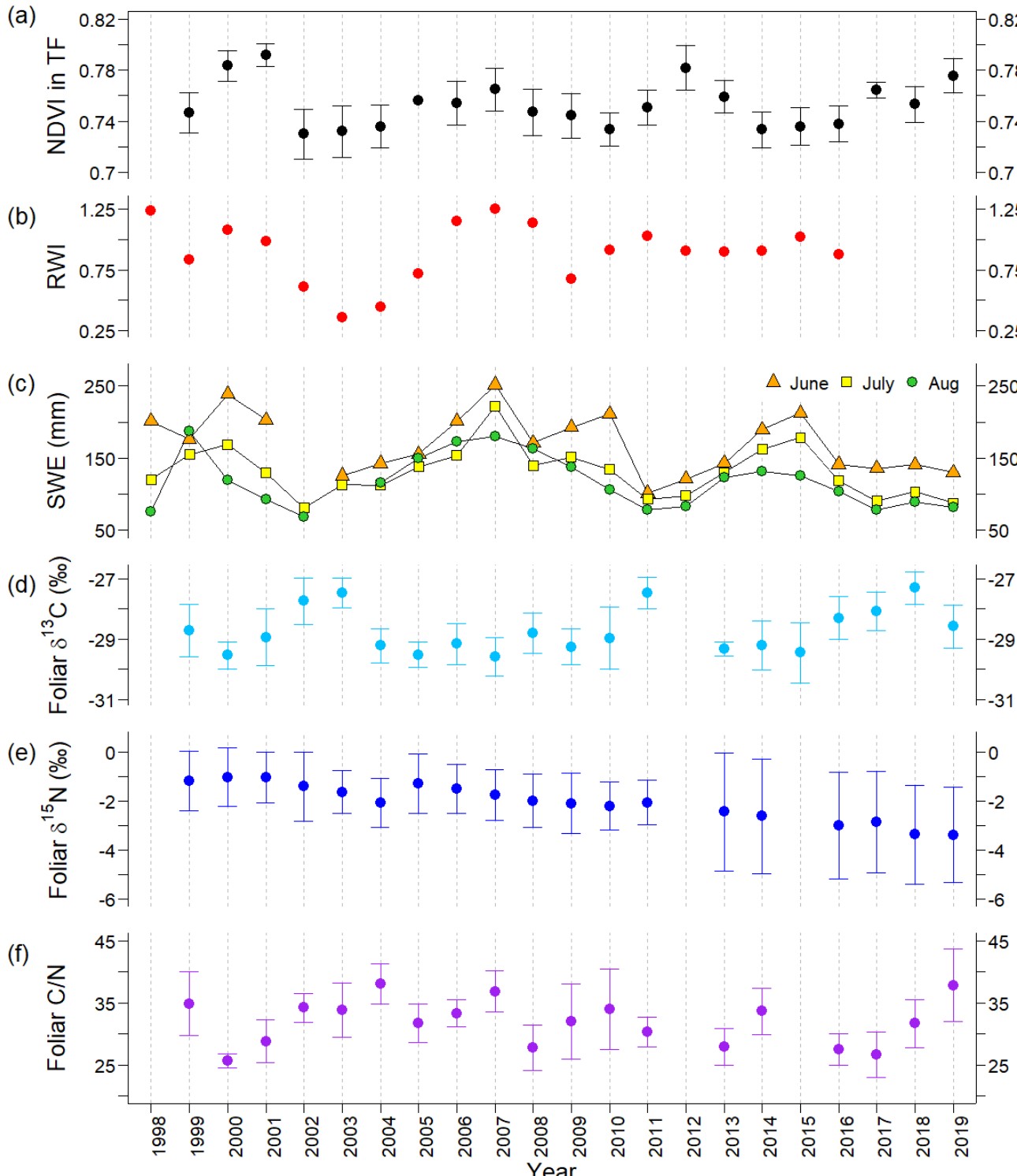

**Figure 3.** The temporal variations in ecosystem parameters observed during 1998–2019 at the typical forest (TF): (a) NDVI, (b) larch ring width index (RWI), (c) soil moisture water equivalent (SWE) at the depth of 0–60 cm in June, July, and August, (d) average foliar $\delta^{13}$C, (e) $\delta^{15}$N, and (f) C/N ratio. Error bars represent standard deviations. There were no data for NDVI in 1998, RWI during 2017–2019, June SWE in 2002, August SWE in 2003, foliar $\delta^{13}$C in 1998 and 2012, and foliar $\delta^{15}$N and C/N in 1998, 2012, and 2015.

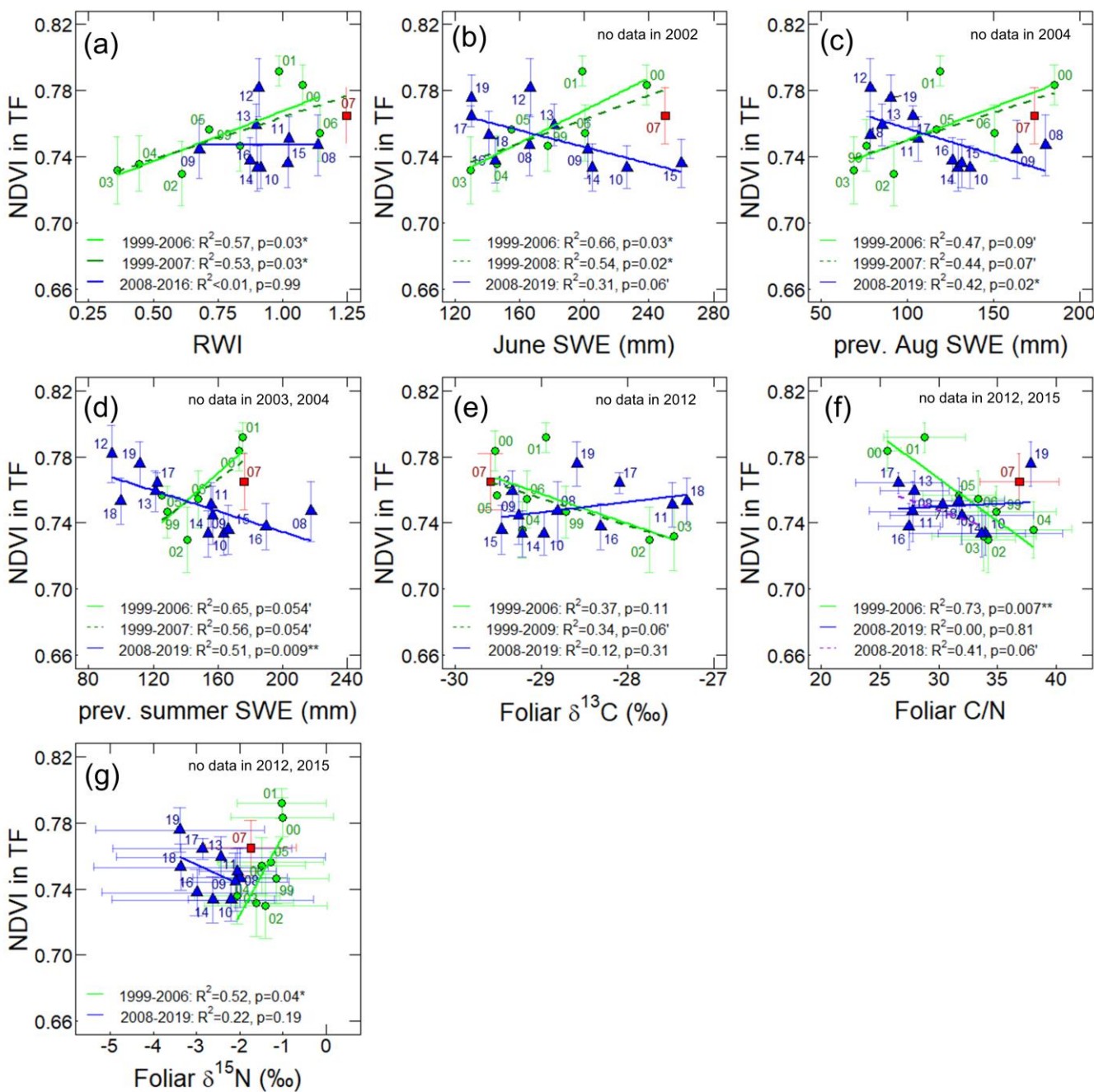

**Figure 4.** The relationships between the TF NDVI in transect and (a) larch RWI during 1999–2016, the monthly average of SWE (mm) in (b) June and (c) the previous August, (d) averaged monthly SWE for June–August of the previous year, (e) foliar $\delta^{13}$C, (f) C/N, and (g) $\delta^{15}$N during 1999–2019. The green circles, red square, and blue triangles show data points during 1999–2006, 2007, and 2008–2019, respectively. Labels nearby the data points are observation years of the TF NDVI. Horizontal and vertical error bars represent standard deviations. Green and blue solid lines show linear regressions for 1999–2006 (before the wet event) and 2008–2019 (after the wet event), respectively, and dark green and purple dotted lines represent other periods. *p*-values and $R^2$ describe the significance and coefficient of determination of the regression models, respectively.

### 3.3.2 RWI at the typical forest

The larch RWI showed a trend similar to that of the transect TF NDVI during 1999–2007 (Fig. 3a and b). The RWI and

average TF NDVI showed high values in 2000–2001 (0.98–1.08 and 0.78–0.79) followed by low values in 2002–2003 (0.36–0.61 and 0.73); the RWI in 2003 was the lowest for the whole observation period. Subsequently, both parameters increased by 2007 (1.25 and 0.76). After 2007, these two parameters exhibited different behaviors. During the period from 2010 to 2013, a one-year time lag was observed in the TF NDVI: there was an increase in RWI from 2009 to 2011, a decrease in 2012, and one year later, from 2010 to 2012, the TF NDVI increased and then decreased in 2013. Statistically, the temporal correlation between the TF NDVI and RWI was positive at a significant level during 1999–2016 ($r = 0.47$, $p < 0.05$; Table S7), with a stronger significant positive correlation before 2007 ($r = 0.76$, $p < 0.05$; Fig. 4a, Table S5) and an insignificant negative correlation after 2007 (Fig. 4a, Table S6).

### 3.3.3 Soil moisture water equivalent at the typical forest

The time series of the SWE and TF NDVI showed different correlations in the early and late halves of the observation period (Fig. 3a and 3c). During 1999–2007, the averaged SWE for June–August (hereafter, summer SWE) and the TF NDVI mostly showed similar trends. High values of the TF NDVI in 2000 and 2001 corresponded to high values of the SWE in the current June (239 and 202 mm) and in the last summer (173 and 176 mm in 1999 and 2000). These high values of TF NDVI and SWE were followed by low values during the drought period in 2002–2003. Subsequently, as summer SWE increased from 2004 (124 mm) to 2007 (218 mm), the TF NDVI also increased. For the period from 2008 to 2019, the correlation between the TF NDVI and summer SWE was negative, with a one-year time lag in the SWE (Fig. 3a and 3c). A low summer SWE value was observed in 2011 (91 mm), and a high TF NDVI value was observed in the subsequent year, 2012. After 2016, the TF NDVI showed an increasing trend, whereas the SWE decreased from 2015 to 2019. Statistically, the TF NDVI showed positive correlations with the SWE in the current June ($r = 0.83$, $p < 0.05$) and the previous summer ($r = 0.79$, $p < 0.1$), including the previous July ($r = 0.82$, $p < 0.05$) and previous August ($r = 0.69$, $p < 0.1$), during the period from 1999 to 2006 (Fig. 4b–d and S4d; Table S5). However, after 2008, TF NDVI showed negative correlations with the SWE in the previous ($r = -0.65$, $p < 0.05$) and current ($r = -0.73$, $p < 0.01$) summer, at a stronger significance level (Fig. 4b–d and S4a–e; Table S6). During and after 2007, there was no change in the TF NDVI; slightly damaged RF-1 showed a decrease in NDVI to levels similar to those observed during the 2002 drought (Fig. 2b).

### 3.3.4 Larch needle δ¹³C, δ¹⁵N, C/N at the typical forest

As shown in Fig. 3a and 3d, the foliar δ¹³C and TF NDVI moved in opposite directions in the early half of the observation period (from 1999 to 2009) (Fig. 3a and 3d). For example, in 2000 and 2001, the TF NDVI had large values, while the foliar δ¹³C values were low (-29.5 ± 0.5 and -28.9 ± 0.9‰). During the period from 2002 to 2007, TF NDVI increased, and at the same time, foliar δ¹³C values decreased. Foliar δ¹³C values higher than -28.0‰ were observed in 2002, 2003, 2011, and 2018, when low summer SWE and TF NDVI were observed (Fig. 3a and 3d). The correlation between foliar δ¹³C and TF NDVI was statistically insignificant without a time lag (Fig. 3e, Tables S5–7), but there was a significant correlation between foliar δ¹³C and TF NDVI with a one-year time lag of foliar δ¹³C after the wet event (Table S6). Foliar δ¹³C was negatively correlated with the previous August SWE during 1999–2007 ($r$ = -0.79, $p$ < 0.05) and 1999–2019 ($r$ = -0.62, $p$ < 0.01 (Fig. 5), but also with the SWE in the current year June and July for the period from 1999 to 2007, and June to August for 2008 to 2019 (Table S8).

Similar to δ¹³C, foliar C/N and TF NDVI moved in opposite directions (Fig. 3a and 3f). In 2000 and 2001, the foliar C/N had low values (25.6 ± 1.1 and 28.8 ± 3.4, respectively), while the TF NDVI was high. There was also a distinct negative correlation between trends in C/N and TF NDVI before and after 2007, excluding 2019 (Fig. 4f).

Foliar δ¹⁵N values decreased after 2005 (Figure 3e). A positive correlation was observed between foliar δ¹⁵N and TF NDVI before 2007 (Fig. 3a, 3f, and 4g).

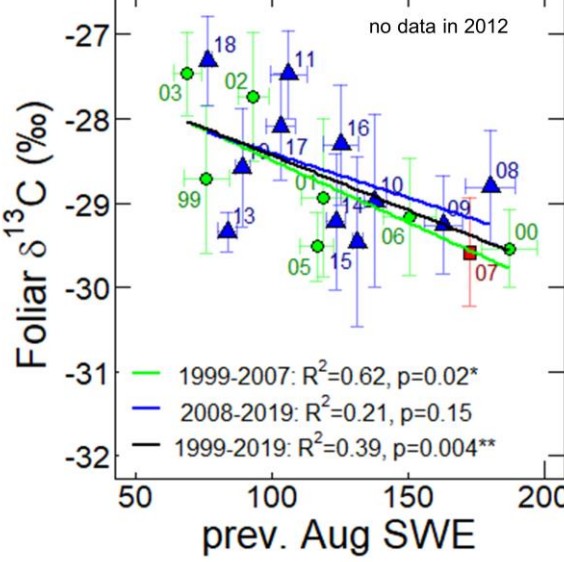

**Figure 5.** The relationship between the foliar δ¹³C and monthly mean SWE in the previous August (mm) during 1999–2019. The green circles, red square, and blue triangles show data points during 1999–2006, 2007, and 2008–2019, respectively. Labels nearby the data points are observation years of the foliar δ¹³C. Vertical error bars represent standard deviations. The linear regressions for 1999–2007, 2008–2019, and 1999–2019 are presented by green, blue, and black solid lines. $p$-values and $R^2$ describe the significance and coefficient of determination of the regression models, respectively.

**4 Discussion**

**4.1 NDVI variation among forest conditions**

Before 2007, there was a small difference in the NDVI among the four forest types (Fig. 2b and Table S3 and S4). In some years before 2007, the NDVI values in RF and DF were higher than those in TF. The difference in elevation between the south and north ends of the transect (approximately 5 m according to Google Earth) may lead to differences in soil moisture; therefore, the RF and DF plots showed higher soil moisture contents and a lower possibility of drought compared with TF before 2007. During 2007–2008, there was a large difference in NDVI among the forest types, especially between the TF and DF (Fig. 2a). During this period, the SWE reached extremely high values (Fig. 3c), caused by a large precipitation amount during 2005–2008 (Fig. 2d). Consequently, the forest floor was partially waterlogged, resulting in damage to the larch forest, especially in the DF and RF plots. These data indicate that during the drought years (before 2007), wet sites such as DF and RF showed higher NDVI values than dry TF sites because of higher water availability. However, after 2007, the TF, which was visually unaffected by the wet event, showed a higher NDVI than the DF and RF. The presence of surface water in DF and soil saturated with water in DF and RF could also reduce the NDVI values. Water predominantly absorbs NIR radiation and therefore has a low NIR reflectance, resulting in a lower NDVI than that of vegetation (Holben, 1986).

After 2009, as the soil became dry, the difference in NDVI among the forest types decreased (Fig. 2b). This may have been caused by the change in the vegetation in RF and DF, that is, the change from mature larch trees to understory and floor vegetation, such as water-tolerant species and seedlings of birch and larch trees via secondary succession. In 2016, the difference in NDVI between TF and DF increased again (Fig. 2b). This may have been caused by the high SWE observed in 2015 (Fig. 3c), which lowered the NDVI in RF and DF.

However, the difference in NDVI between TF and DF remained at the end of the observation period. Previously, the spatial variation in NDVI along the transect was investigated a decade after the wet event in 2018 (Nogovitcyn et al., 2022). Nogovitcyn et al. (2022) concluded that NDVI was higher in TF than in DF because of a difference in the stand density of mature trees, as NDVI indicates a leaf area index (LAI), which corresponds to the number of mature trees in this forest.

As described in the Methods 2.2, we combined images from three Landsat satellites with different sensors. Although combining data from different sensors can lead to uncertainty in the signals (Shin et al., 2023), our result or historical variation in the NDVI reflected the change in the forest condition observed *in situ*.

**4.2 Trends in NDVI of the transect and 10-km plot**

The NDVI of the 10-km plot showed a trend similar to that of the transect NDVI during the observation period ($r = 0.78$, $p < 0.001$), as shown in Fig. 2a, and the mean NDVI value of the 10-km plot was lower than that of the transect in most years. We found year-to-year variations in both NDVI datasets, but no significant increasing or decreasing trends were observed,

which is consistent with the observations during previous studies at this site (Nagano et al., 2022; Tei et al., 2019b; Lloyd et al., 2011). Therefore, our observational data can be used for the analyses of ecosystem changes not only at the plot scale but also at the regional scale.

### 4.3 Historical variation in NDVI of typical forest

We studied historical variations in NDVI and field-observed ecosystem and climatic parameters of a typical forest to understand forest conditions.

### 4.3.1 Water availability

As described in the Sect. 3.3.3, SWE controls forest NDVI because the observation site (northeastern taiga) is established in a continental dry area. We found positive and negative correlations between the NDVI and SWE. Before 2007, the TF NDVI was positively correlated with the June SWE in the current year (Fig. 4b) and positively correlated with the SWE in the previous year June, July, August, and the previous year summer (JJA: June–July–August) (Fig. 4c, 4d, S4c, and S4d, Table S5). This indicates the influence of hydrological conditions in the previous year and early summer of the current year on the leaf productivity of larch trees in the current year.

Larches, as deciduous trees, assimilate carbon through photosynthesis (photoassimilate) during the summer to prepare needles in the next year, and the elongation of needles may be affected by hydrological conditions in the early summer. In the Spasskaya Pad Forest, pulse-labeling experiments with $^{13}CO_2$ showed that stored carbon from the previous year contributed approximately 50 % to formation of new needles in *Larix gmelini* saplings (Kagawa et al., 2006). The high level of water availability in the summers of 1999 and 2000 likely contributed to increased carbon storage and, as a result, the high formation of needles in 2000 and 2001. The significant NDVI decrease in 2002 was probably caused by a low level of soil moisture (i.e., dry conditions). The high summer air temperature (Fig. 2c) and the small amount of precipitation (Fig. 2d) in 2001 and 2002 caused droughts in 2002 and 2003. Subsequently, the soil moisture increased due to a large amount of water year precipitation (Fig. 2d), which likely contributed to an increase in NDVI until 2007.

It is known that the NDVI depends on the previous-year precipitation in arid and semi-arid regions (e.g., Burry et al., 2018; Camberlin et al., 2007). In addition, historical time series of climate indices, based on both precipitation and temperature, were related to one-year lagged NDVI (e.g., Verbyla, 2015; Liu et al., 2017). In boreal interior Alaska, the summer moisture index showed a correlation with maximum summer NDVI not only at a one-year time lag in two 10-km climate station buffers but also at a two-year time lag in many other ones (Verbyla, 2015). Possible reasons for the multi-year NDVI lag could be the long-term negative vegetation responses to drought events, such as a decrease in carbon allocation by plants (e.g., Kannenberg et al., 2019) and plant mortality (e.g., Anderegg et al., 2012). Negative effects of drought events also occurred in our study.

As described above, the positive correlations between the TF NDVI and soil moisture were observed during 1999–2006;
however, the correlations were shifted to negative values during 2008–2019 (Fig. 4b–d and S4a–e). After 2007, the TF NDVI
was negatively correlated with the SWE of all months in the previous (with a one-year time lag) and current years (without a
lag) (Table S6). This may indicate that after the extreme wet event, the soil moisture in the previous and current years seemed
to negatively affect the current TF NDVI. Therefore, a high level of soil moisture may affect needle production (i.e., carbon
assimilation, needle formation, and/or needle elongation).
Additionally, the $\delta^{13}C$ values of needles at our study site often depend on water availability (Kagawa et al., 2003; Tei et al.,
2019a). As shown in Fig. 5, there was the significant negative correlation between foliar $\delta^{13}C$ and the previous August SWE
in 1999–2007 ($r = -0.79$, $p < 0.05$). Interestingly, not only before the wet event but also for the whole observation period
(1999-2019), a negative correlation was found between foliar $\delta^{13}C$ and the previous August SWE (Fig. 5). These results
differed from the correlations between the NDVI and SWE, which changed from positive to negative values. Before the wet
event, under drought stress during 2001–2002, needle stomatal conductance was decreased, resulting in decreased carbon
assimilation. In the subsequent years, 2002–2003, larches probably produced fewer needles (lower NDVI) with higher $\delta^{13}C$
from the previously photosynthesized C (Fig. 3a and 3d).
After the wet event, the foliar $\delta^{13}C$ and SWE remained negative, indicating high stomatal conductance (low foliar $\delta^{13}C$). High
stomatal conductance typically contributes to the higher potential of a plant to assimilate $CO_2$, store C, and produce needles
(high TF NDVI); however, after the wet period, larch produced fewer needles (low TF NDVI).
Compared with the decrease in the TF NDVI for drought events, that for the extreme wet event was smaller (Fig. 2b and 3a),
although the extreme wet event caused a significant decrease in the NDVI of RF-1 and RF-2. The decrease in the TF NDVI
in wet years may be due to different factors, such as nitrogen availability for larches, which can control needle formation.
This factor will be discussed in the next chapter.

### 4.3.2 Nitrogen availability

Before 2007, the TF NDVI showed a significant negative correlation with foliar C/N (Fig. 4f), indicating a positive
correlation with foliar N content. In this ecosystem, there have been no previous studies on the temporal correlation between
NDVI and plant N content (or $\delta^{15}N$). Changes in leaf nitrogen, which is an important element of chlorophyll (green pigment),
were detected using NDVI (Gamon et al., 1995). Previously, the relationship between NDVI and leaf N content was
predominantly investigated in crops for agricultural purposes but not in natural ecosystems. In coniferous forests, the
estimation of foliar nitrogen using remote-sensing methods showed the highest uncertainty due to the complex structure of
needleleaf canopies (reviewed by Homolova et al., 2013).
As leaf N content is considered to be an indicator of nitrogen availability for a plant in some boreal regions, where the
ecosystem is usually poor in N (Matsushima et al., 2012; Liang et al., 2014), we concluded that forest greenness (NDVI) was

strongly controlled by nitrogen uptake by larch trees. Therefore, soil moisture is suggested to play a crucial role in maintaining forest nitrogen status. During 2000–2001, soil water was available for plants and induced favorable conditions for soil nitrogen uptake by trees. Under suitable soil moisture conditions, the production of soil inorganic N may increase. This may lead to a high production of larch needles (high NDVI). During 2002–2003—the drought years—dry conditions caused less productivity of soil inorganic N and less N uptake by trees. In the post-drought period of 2004–2007, an increase in soil moisture gradually recovered the forest conditions in terms of nitrogen uptake and needle production.

After 2007, the foliar C/N still showed a negative correlation with the TF NDVI during 2008–2018, but this correlation was statistically weaker compared to that during 1999–2006 (Fig. 4f). At the same time, the positive correlations between the TF NDVI and SWE changed to negative ones during 2008–2019. According to these results, high soil moisture could lead to low needle production under low nitrogen availability. When extremely high soil moisture, resulting in the saturation of soil with water, caused less production of soil inorganic nitrogen, low TF NDVI and high C/N values may be observed; thus, TF NDVI and SWE were negatively correlated.

While N content reflects the plant's nitrogen status, plant $\delta^{15}N$ is widely accepted to depend on the isotopic composition of nitrogen sources (e.g., Evans, 2001). Therefore, the $\delta^{15}N$ of soil inorganic ammonium $NH_4^+$, which is the main nitrogen source in the Spasskaya Pad forest (Popova et al., 2013), presumably determined the foliar $\delta^{15}N$ in larches. As shown in Fig. 3e, foliar $\delta^{15}N$ gradually decreased after 2005. These data suggest that larch trees used less soil inorganic N, especially from the deeper soil layers, which usually have higher soil $\delta^{15}N$ (Amundson et al., 2003; Fujiyoshi et al., 2017). This may be related to either a change in soil N dynamics, a decrease in the vertical distribution of roots (Takenaka et al., 2016), or damage to the lower roots due to extremely high soil moisture. Under root oxygen stress due to soil flooding, plant metabolism changes from aerobic respiration to anaerobic fermentation, characterized by energy deficiency and ethanol production, both of which induce decreased nutrient uptake and plant growth (reviewed by Pezeshki and Delaune, 2012). Reduced soil conditions can also induce soil phytotoxin production, damaging the root system (Pezeshki, 2001).

It should be noted that not only the extreme wet event in 2007, but also the extreme drought in 2001 may have caused a change in N availability. Many studies have shown that foliar $\delta^{15}N$ increases during drought (Penuelas et al., 2000; Handley et al., 1999; Lopes and Araus, 2006; Ogaya and Penuelas, 2008). However, in the present study, the drought in 2001 and 2002 decreased foliar $\delta^{15}N$. We could not identify the exact reason, but drought in 2001 and 2002 might have affected the N availability for larch trees.

### 4.3.3 NDVI and RWI of larch trees

Two parameters of aboveground biomass, the RWI and TF NDVI, were positively correlated at a significant level ($r = 0.76$, $p < 0.05$; Fig. 4a) during 1999–2006. Similarly, in other northern regions, temporal patterns of the NDVI and dendrochronological data were similar for larch (Erasmi et al., 2021; Berner et al., 2011; Berner et al., 2013), pine (Berner et

al., 2011), and spruce (Andreu-Hayles et al., 2011; Beck et al., 2013; Berner et al., 2011; Lopatin et al., 2006). This means that tree growth (RWI) and needle production (NDVI as an indicator of LAI) showed synchronous responses to environmental changes before 2007. However, there was no significant correlation between the TF NDVI and RWI after 2007. Thus, the extreme wet event in 2007 could have changed the physiological response of larch trees to the environment in terms of needle and wood production.

The correlation between NDVI and RWI at our observation site was previously reported by Tei et al. (2019b). They used GIMMS-NDVI3g and found its positive correlation with the RWI in the subsequent year during 2004–2014 at the study site. These two parameters, the NDVI and RWI, reflect the carry-over of carbon, which is fixed via needles in the previous year and used in the current year, as experimentally demonstrated by Kagawa et al. (2006). In previous studies, dendrochronological data showed that tree growth responded to climate with a time lag (e.g., Tei and Sugimoto, 2018). In our study, we did not observe a significant correlation between the TF NDVI and RWI at the one-year lag of RWI (Fig. S4g).

**4.4 Changes in larch forest NDVI due to drought and the extreme wet event**

As shown in Fig. 2b, the NDVI in TF showed a significant decrease in 2002. Such a decrease in NDVI has been repeated in the past because the climate is continental (dry) in this region. Compared to this decrease in NDVI due to drought, the extreme wet event in 2007 showed only a slight decrease in the NDVI in TF, although the NDVI in DF and RF-2 decreased considerably.

Tree mortality in the DF and RF during the extreme wet event is controlled by soil properties and topographic features, that is, depressions (Iwasaki et al., 2010). However, the effects of the event may not only include tree mortality but also invisible damage to living trees. In this study, NDVI, a potential indicator of needle production in a typical forest, was negatively related to the summer SWE in the previous and current years during 2008–2019 and with the current-year needle C/N during 2008–2018 (Table S6). This may indicate that needle production in the current and subsequent years during the summer was disturbed by increased soil moisture and decreased soil N uptake by trees. We suggest that N uptake by larches might be reduced in wet soils due to damaged lower roots, decreased vertical distribution of roots (Takenaka et al., 2016), or altered soil N production.

Changes in the process of needle production may affect tree growth in the current and subsequent years in the forest (Kagawa et al., 2006; Tei et al., 2019b). However, we found no evidence that tree radial growth was disturbed after 2007, and the RWI responded well to changes in the SWE (Fig. 3b and 3c). Additionally, at the ecosystem scale, there was no significant change in the $CO_2$ exchange measured by the 32-m flux tower in this larch forest (Kotani et al., 2019). However, the observed increase in understory biomass was suggested to compensate for negative changes in fluxes at the overstory level (Kotani et al., 2019) and in the NDVI of the forest (Nagano et al., 2022). This means that the negative effects of the extreme wet event on living larch trees were not excluded in the previous studies. Our study showed that limitation in N uptake at high soil

moisture levels is one of the factors that may potentially reduce tree growth in the future.
In our previous study (Nogovitcyn et al., 2022), spatial variations in NDVI and foliar traits identified favorable conditions in
the sites affected by the extreme wet event (RF). The larch forest in RF with lower NDVI (lower stand density) had higher
light (higher $\delta^{13}$C) and nitrogen (lower foliar C/N) availability for one mature larch tree than that in unaffected areas (TF)
because of reduced competition for light and soil nitrogen among trees. Such favorable conditions and the presence of a large
number of young larch trees may lead to further RF succession after an extremely wet event. However, because weather
extremes are expected to be more frequent and intensive (Douville et al., 2021), the period between extremes may exceed the
period of recovery after the extremes. Therefore, in the future, forests may be damaged rather than recovered.
We showed that NDVI values in affected areas were lower than those in typical larch forests for more than 10 years after the
extreme wet event. In other boreal regions, the NDVI was higher in damaged forests because of the strong contribution of
understory to surface greenness (Bunn et al., 2007; Dearborn and Baltzer, 2021), suggesting the need to combine field and
remote sensing observations. Regarding the prediction of tree growth in this dry region, there is a discrepancy between the
different vegetation models. The dynamic global vegetation model (DGVM) simulated increased forest production
throughout the circumboreal region in the nearest century, whereas the RWI-based model showed the opposite result in the
regions of Alaska, Canada, Europe and our study site (Tei et al., 2017). In these regions with a continentally dry climate, tree
growths suffers because of the effects of temperature-induced drought stress and are predicted to decrease under warming
conditions (Tei et al., 2017). In addition, extreme wet events are also likely to have a negative impact on forest production
because of changes in nitrogen availability, as demonstrated in this study. Some models may overestimate production
because they do not include important parameters such as soil moisture, soil N production, and N uptake by trees. To better
understand changes in the forest, long-term observation of variations in soil N availability depending on soil moisture and
other factors is necessary.
**5 Conclusions**
In this study, historical variations in satellite-derived NDVI (seasonal maximum) and field-observed parameters of larch
forests were investigated to understand the effects of the extreme wet event on the larch forests of northeastern Siberia. The
NDVI values of the plots visually unaffected (typical mature larch forest, TF) and affected by the event were similar before
2007, and the NDVI values at both plots were similarly decreased by drought. However, both NDVI values ~~but~~ differed after
2007 because of the high tree mortality in affected plots caused by waterlogging and the presence of water in the depression.
Although the TF was visually unaffected by the event, it also changed. Before the wet event, the positive relationship
between the TF NDVI and SWE in the previous summer and the current June showed that needle production increased with
water availability, as previously observed in this dry region. However, after the wet event, the relationship between the TF

NDVI and soil moisture in the previous and current years unexpectedly shifted from positive to negative, which may have been related to N availability.

N is considered an important factor controlling needle production before and after the wet event, as the negative correlations between TF NDVI and needle C/N ratio were observed until 2018 (except for in 2007). In addition, the needle $\delta^{15}$N continuously decreased after the wet event, suggesting that the larch trees used a different N source when the lower roots were damaged by anaerobic conditions. Before the wet event, the high (but suitable) soil moisture level presumably produced more soil inorganic N, and consequently produced more larch needles, whereas the extreme wetness after 2007 likely had a long-term negative effect on needle production because of the lower soil N production. As shown in this study, extreme wet events in continental dry regions may alter the interaction between water availability and tree performance (for example, NDVI) over a long time because of shifts in N availability for trees.

**Data availability.** Yakutsk air temperature and precipitation data are available from the RIHMI-WDC website (http://aisori-m. meteo. ru/). NDVI data was calculated from Landsat 5, 7, 8 images, which are available from the USGS website (https://earthexplorer.usgs.gov/). The seasonal maximum NDVI data and larch RWI data are described in the Supplement (Table S1 and S2). All other data presented in this work have been deposited in the Arctic Data Archive System (ADS): soil moisture data at Spasskaya Pad (https://ads.nipr.ac.jp/dataset/A20230217-001) and larch needle C/N and carbon and nitrogen isotopes at Spasskaya Pad (https://ads.nipr.ac.jp/dataset/A20230217-002).

**Author contributions.** AS designed the research and AN calculated the NDVI and performed the analyses. TM, ST, and NS helped with the analyses, and TCM managed all the field observations. RS and YM helped with field observations. AN and AS prepared the paper with contributions from all co-authors.

**Competing interests.** The authors declare that they have no conflict of interest.

**Acknowledgements.** The authors are grateful to Dr. A. Kononov, R. Petrov, and other colleagues from the IBPC for supporting our fieldwork at the Spasskaya Pad Forest Station and M. Grigorev for his assistance in the fieldwork. The authors also appreciate Y. Hoshino, S. Nunohashi, A. Alekseeva, and E. Starostin for their support in laboratory work and logistics.

**Financial support.** This work was supported by the Belmont Forum Arctic program COPERA (C budget of ecosystems, cities, and villages on permafrost in the eastern Russian Arctic) project, the International Priority Graduate Programs (IPGP), funded by the Ministry of Education, Culture, Sports, Science, and Technology-Japan (MEXT), and the Hokkaido University DX Doctoral Fellowship (Grant No. JPMJSP2119), funded by the Japan Science and Technology Agency.

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
