# Peer review of "Historical variation in normalized difference vegetation index"

_EGUsphere, 2023_

## Referee Comment (RC2)

**Review:**

**Nogovitcyn, A., Shakhmatov, R., Morozumi, T., Tei, S., Miyamoto, Y., Shin, N., Maximov, T. C., and**
**Sugimoto, A.: Historical variation in normalized difference vegetation index compared with soil**
**moisture at a taiga forest ecosystem in northeastern Siberia, EGUsphere [preprint],**
**https://doi.org/10.5194/egusphere-2023-279, 2023.**

In "Historical variation in normalized difference vegetation index compared with soil moisture at a taiga forest ecosystem in northeastern Siberia" the authors investigated the variation in NDVI among forest conditions ( typical mature, TF; regenerating-1, RF-1;  regenerating-2, RF-2; and damaged forests, DF) and field-observed parameters (from 1998 to 2019) such as RWI, soil moisture, changes of larch needles ( $\delta$ 13C,  $\delta$ 15N, C/N), air temperature, and precipitation. The authors determined that prior to the 2007 extreme wet event, wet areas like DF and RF had higher NDVI values than dry

TF sites due to greater water availability. However, following 2007, the TF had a greater NDVI than the DF and RF, although being visibly unaffected by the wet event.

Studying historical variations in NDVI compared with soil moisture at a taiga forest ecosystem in north-eastern Siberia is important for several reasons. Firstly, NDVI data can provide valuable information about temporal and spatial changes in vegetation distribution, productivity, and dynamics, which allows for the monitoring of habitat degradation and fragmentation. Secondly, the comparison of historical variations in NDVI with soil moisture can provide insights into the impact of extreme weather events on vegetation, such as the extreme wet event in 2007, which resulted in high tree mortality and a decrease in NDVI at affected sites. Understanding the ecological effects of climatic disasters such as drought or fire can be assessed using NDVI data, making it a valuable tool for monitoring changes in vegetation due to climate change. Overall, studying historical variations in

NDVI and soil moisture in a taiga forest ecosystem can provide valuable insights into the impact of extreme weather events on vegetation and the effects of climate change on vegetation dynamics.

Therefore, this paper has the potential to make an important contribution to the body of knowledge concerning the impacts of global change on sensitive and complex permafrost ecosystems.

It is my opinion that the authors used sound methods to address the study aims and presented the research findings clearly and concisely and they used appropriate figures to illustrate the NDVI values of the forest types and the trends in the transect and 10-km plot, which could be useful for researchers and policymakers. However, I agree with referee 1 about their main points raised as well as the minor comments provided. To avoid repetition and in the interest of brevity, I will not be going over them again in this review, but I strongly advise the authors to make the corrections already suggested. Instead, I will just add a few points concerning the discussion section that I would like to see addressed before publication. When the authors revise these issues, I recommend the study for publication in Biogeosciences.

In the discussion, the authors considered the probable reasons for the differences in NDVI values among the forest types, such as the change in vegetation and the presence of surface water and saturated soil. However, the section could benefit from a more critical evaluation of the results and their implications. For example, the article does not address the limitations of using NDVI as a proxy for vegetation health and productivity, which could impact the accuracy of the results. NDVI

measures the amount of chlorophyll in the uppermost layers of vegetation. This means that it may not accurately represent the health and productivity of plants with lower canopies or those that are hidden from view. The limitations of using NDVI as a proxy for vegetation health and productivity may be particularly relevant in taiga/permafrost ecosystems due to their complex vegetation structure and sensitivity to environmental changes.

Additionally, the article does not explore the broader ecological implications of these findings, such as how changes in vegetation health and productivity may impact ecosystem services or the ability of forests to sequester carbon. Finally, while the article notes the potential for using the observational data for analyses of ecosystem changes at the plot and regional scales, it does not explicitly state what these analyses might entail or why they would be valuable. A more explicit discussion of the practical applications of the research could make the findings more accessible to a wider audience.

---

## Author Comment (AC1)

Response to Reviewer 1 Comments.

Thank you very much for your valuable comments. We numbered all comments and replied to them. According to your comments, we will revise our manuscript. We have replied to each of them, and if our proposed revisions are clear and sufficient for you, we will modify the manuscript according to them. For some comments, we showed the revisions, but before we submit the revise manuscript, the text will be edited by native speaker, although there is no change in the contents.

Answers to the main comments:

(1) In the introduction, discussion and particularly the conclusion, the authors mostly discuss earlier findings from Spasskaya Pad, and hardly touch upon potential similarities or dissimilarities with other regions. This makes it very hard for the reader to assess to what extent the findings presented here may hold lessons for the other boreal forests on permafrost. In my view, your results hold important lessons for the potential impacts of increased precipitation variability in northern forests, also beyond Siberian larch forests! Precipitation variability is increasing rapidly in this region (see also https://doi.org/10.1016/j.jhydrol.2021.126865) so it is important to discuss what your findings imply for the future functioning of Siberian larch forests and potentially boreal forests in general. You also demonstrate a clear "legacy effect" that could be related to recent insights regarding duration of the impacts of extremes (see for instance https://onlinelibrary.wiley.com/doi/pdfdirect/10.1111/gcb.16078). You still find divergence in NDVI over ten years after an extreme event. This is a major legacy effect, that has important implications for knowledge on Arctic greening/browning and should be stressed more strongly in the conclusion and abstract!

We will add explanations about the similarities or dissimilarities with other regions in introduction, 4.4, and conclusions. Extreme weather events such as heavy precipitation and snowfall will affect the ecosystem change over the long term. We will put these to our manuscript.

(2) The described aim of the research is to assess how the local forest has changed over time, but throughout the methods you have decided a priori to split up the data into a pre-2007 and post-2007 segment based on an extreme event. Hence, it seems more appropriate to either first statistically evaluate and demonstrate whether there is a significant trend break. I do not doubt this would be the case if you would try it, but it would provide a back up for your methodological choice. Alternatively (maybe this is easier) you could reframe the research aim to explicitly investigate the effect of this wet event. This would make sense, since the subdivision of forest types within the transect seems to already be based on forest damage and

regeneration stadia, and the introduction already extensively discussed observed effects of the 2007 wet period.

Yes, I agree with your comment. Our study is not only the extreme wet event, and I would like to show the historical variation of the larch forest using NDVI. The larch forest at our study and also northeastern Siberian taiga site have been suffered from drought and recently wet event. We would like to show how drought and wet event affect the NDVI. But for the most visible and impacted change in the correlation between NDVI and ecosystem parameter (especially soil moisture) was the wet event. It is not possible to change the statistical analysis at this moment, therefore we will reframe the research aim to investigate the effect of wet event as described by the reviewer.

(3) The ecophysiological meaning of the d15N, d13C and C/N ratio data, as well as the methods through which they were derived, are completely lacking. The reader will need more background to understand the presented patterns and the methods are not reproducible here.

We added the explanations. Read the replies to the comments (15), (16), (26).

(4) I have some concerns about confounding effects of seasonal availability of landsat ndvi data in shaping the temporal dynamics of ndvi and affecting relationships with other site data. In the line comments, I have added some examples and suggestions on how to deal with this. I think with an additional figure or potentially addition of covariates/interactions such issues could be resolved quite well.

When we use the satellite image data, there are many problems such as temporal resolution and combination of different sensors. Please read the replies to the comments (14), (19), and (20). We tried to make as much as possible.

Answers to the Line comments:

(5) L. 29: Could you reflect briefly on the implications of your results to place them in a wider context? Parts of the Siberian Arctic show record browning in recent decades, as you undoubtedly know better than anyone. Perhaps you could reflect on the potential role of moisture dynamics, drought and waterlogging in this browning trend? (Just a suggestion).

Yes. As you know, boreal forests in northeastern Siberia are experiencing browning, because of not only by temperature-induced drought but also waterlogging and nitrogen dynamics as we showed in our manuscript. We will add the sentences to the abstract.

Revision: "... due to extremely high soil moisture. Taiga in northeastern Siberia, as a dry forest ecosystem, is experiencing browning because of not only temperature-induced drought but also waterlogging and nitrogen dynamics."

(6) L. 31-32 "occupy a large forest area, approximately 27 % (Fao, 2020)" --> I assume you mean 27% of the world's forest cover? Could you rewrite this to make it clearer what the statistic refers to? Also consider writing "FAO" instead of "Fao" as you also write it in the reference list.

Thank you very much. We changed Fao to FAO. According to FAO, 27% is the percentage of boreal forest in the total forest cover.

Revision: "...occupy a large  area, approximately 27 % of the world's forest cover ( FAO, 2020)".

(7) L. 39 "and change the ecosystem" --> Could you provide a few concise examples and references?

Revision: "Under warming, permafrost may decline, which can trigger large amounts of carbon emissions (Schuur et al., 2015) contributing to further climate warming, alter soil hydrology (Walvoord and Kurylyk, 2016), soil nutrient availability (Salmon et al., 2016), vegetation cover (Jin et al., 2021) and other, and thus change the ecosystem."

(8) L. 31 - 66: Please consider adding some thematic structuring to the introduction; the introduction seems to give an overview of earlier work that is mostly focused on C-exchange, while the knowledge gap decsribed on L. 65-66 focuses on NDVI and foliar parameters.

We will add the explanations about the knowledge gap. We will also add the characteristics of our study site and comparing with other regions.

(9) L. 67 - 70: The research aim is described as "assessing how the forest has changed", which seems unnecessarily vague. Could you provide more specific aims or research questions and (optionally) hypotheses? Setting more specific aims may also help provide structure and direction to the introduction paragraph above.

The sentence in L. 68-69 was described the outline of our aim, and this looks vague. We changed the paragraph as below.

Revision: "The purpose of this study was to understand how the larch forest in this region changed and what factors impacted the changes over the past two decades.  To do these, we investigated historical variations in satellite-derived NDVI to know the change of forest, because leaves of deciduous tree reflect the condition of tree in every year. In order to understand the factors, which impact the forest change, NDVI data were compared with  field-observed parameters, such as the RWI, soil moisture, needle $\delta^{13}$C, $\delta^{15}$N, C/N, air temperature, and precipitation from 1998 to 2019."

(10) L. 78: "consists of deciduous species" --> any information which ones? do they occupy a significant share of the canopy compared to dominant larch vegetation?

Revisions:

L.78. "Dominant tree species is larch  *(Larix cajanderi)* that is a deciduous conifer (Abaimov et al., 1998), mixed with broadleaved birch (*Betula pendula*), ...".

L. 35. "with coniferous trees" → "with deciduous conifers"

 L. 80 " and other grasses" --> please remove "other" (as the shrubs mentioned before are not grasses)

Revision: "... and  grasses."

(12) L. 95: "Regenerating forests RF-2 had moderate forest conditions between RF-1 and DF" --> what do you mean by moderate forest conditions?

Revision: L. 90-95. "The plots discerned as regenerating forests RF-1, had many dead mature larches and formed forest gaps in the overstory where there were a large number of young larches (seedlings and saplings with a height of up to 3 m) and shrubs. Regenerating forests, RF-2, had more dead mature larches and more young larches than RF-1. Damaged forests, DF, where all mature trees died, were predominantly covered by moisture-tolerant grasses, and had much smaller numbers of young larches than in RF-1 and RF-2. The DF plots were located on a depression in a trough-and-mound topography, and some patches of the DF plots were flooded.  "

(13) L. 108 - 110: " The transect plots, which consist of pixels not attributed to quality pixels (clear terrain, low-confidence cloud, and low-confidence cirrus) in the quality assessment bit index band according to Landsat Surface Reflectance product guides, were excluded from the analysis. --> due to the structure of this sentence it reads to me as though all transect plots ndvi values were excluded from analysis, but as the text continuous you describe how it was used in further analysis, so I assume you only removed pixels (or transect plots?) that were flagged in the QA product? Perhaps you could rephrase this more clearly (e.g. that "pixels flagged in the quality assessment bands were omitted from analysis"? or that "transect plots that contained pixels flagged in the quality assessment bands were omitted from analysis"?).

Revision: "The transect plots, which consist of only pixels  attributed to quality pixels (clear terrain, low-confidence cloud, and low-confidence cirrus) in the quality assessment bit index band according to Landsat Surface Reflectance product guides, were  used in the analysis."

(14) L. 120: can you provide an assessment of fit among the different sensors, e.g. on days for which multiple products are available? how accurate is the estimate for the one sensor based on another sensor compared to the actual value? Roy et al 2016 recommend to use a locally

parameterized regression, although it would be understandable if insufficient overlap in acquisitions among different sensors prevents establishment of specific regression parameters for your site.

We understand that local parameterization is important, because it is not possible to combine different sensors perfectly. However, unfortunately, we cannot show the sufficient data of assessments for publication. In our study, three Landsat images (Landsat 5 TM (L5), Landsat 7 ETM+ (L7), and Landsat 8 OLI (L8)) were available. L7 had the longest observation period, but actually data quality was not so good, compared to L5 and L8 (after the scan-line corrector failure of L7 in 2003). After the selection of image data and conversion by Roy et al. (2016) and Ju and Masek (2016) as described in Methods 2.2, and we again selected the NDVI data for comparisons between L7 and L5 for the period 1999-2011, and L8 and L7 for 2013-2019, by the following conditions.

- For transect plots, all 34 transect plots were observed. For 10-km plot, more than 96% of pixels in L5 and L8, and more than 75% of pixels in L7 were observed.
- There was one day difference in the acquisition dates between L5 and L7 and between L7 and L8, and NDVI signals were close.
- If the average value for the short period in summer (NDVI shows usually small change in July to beginning of August) was calculated, we used the average value.

Eleven data (including transect and 10-km plot) for comparison between L7 and L5 and twelve data (including transect and 10-km plot) between L8 and L7 were identified. The results were close to the 1:1 line (see the figure below).

There are many problems on statistical procedure if we show these assessments in our paper. But we believe that the conversions by Roy et al. (2016) and Ju and Masek (2016) can be used realistically.

We put some sentences to Methods 2.2 L.120 and Discussion 4.1 between L. 291 and L. 292.

[Figure]

Revisions:

L120:"The local parametrization of signals from different sensors was not performed for our study site due to insufficient overlap in the image acquisitions. We identified that the selected and converted data were close to the 1:1 lines between Landsat 5 and 7 and between Landsat 7 and 8".

Between L291 and 292: "As described in the Methods 2.2, we combined images of three Landsat satellites with different sensors. Although there is an uncertainty of the signals by combinations of different sensors (Nagai et al., 2022), our result of historical variation of NDVI reflected the change of forest condition observed in situ."

(15) L. 133-136: this paragraph lacks context of the ecological or physiological meaning of isotope ratios and C/N ratio. More explanation and literature is needed for the non-expert reader to assess what the d15N, d13C, C/N ratios and ring widths actually mean and what questions you are answering by including these data (alternatively, you could also already explain how the different types of datasets relate to the research aims in the final introduction paragraph)

We added some sentences for explanation and literature.

Revisions:

L. 133-135: "Several ecosystem parameters have been observed since 1998 in typical forests. To monitor the physiological response of larch to environmental changes, the carbon and nitrogen isotopic compositions ($\delta^{13}$C (‰), and $\delta^{15}$N, (‰), and the ratio of carbon to nitrogen content (C/N) of larch needles have been observed since 1999, except in 2012, (at the site 0.2km200 m south of the transect; (Fig. 1b). The $\delta^{13}$C and $\delta^{15}$N are calculated by:

$$\delta^{13}C \text{ (or } \delta^{15}N) = (R_{sample}/R_{std} - 1) \times 1000 \ (‰),$$

where R$_{sample}$ and R$_{std}$ are isotope ratios ($^{13}$C/$^{12}$C or $^{15}$N/$^{14}$N) of the sample and standard, respectively, and standards are Vienna Peedee Belemnite (VPDB) for carbon and atmospheric N$_2$ for nitrogen. The foliar $\delta^{13}$C reflects the physiological condition of photosynthesis and has been widely applied to indicate plant water use efficiency (Farquhar et al., 1989). The foliar $\delta^{13}$C becomes high when the higher irradiance and lower stomatal conductance are observed. The foliar $\delta^{15}$N is a physiological indicator of the nitrogen source for a plant (Evans, 2001), which can vary depending on many physiological and environmental factors. The foliar C/N represents the nitrogen status of a plant (Liu et al., 2005). ..."

L. 138-140: "For more than 100 years, until 2016, larch ring-width index (RWI), which indicates wood growth dynamics, was estimated by detrending and standardizing the raw time-series width data obtained from the collected paired cores (Tei et al., 2019b). The RWI data used for analysis are shown in Table S2."

(16) L. 133-136: There seems to be no explanation of how the d15N, d13C and C/N ratios were derived, Add methodology (which tissues were sampled, how many grams, how were they analyzed, on which instrument, against which isotope standards at what precision?). If the data come from an existing dataset or study, please cite it so the reader can understand how the values were derived.

We put the outline of methods in Methods 2.3, and method details were added in the caption of Figure S1.

Revisions:

L. 135-137: " Larch needles were collected from four to eight young larch trees  in August every year. The details of sampling, sample preparation and laboratory analyses for C and N contents and for their isotope compositions using the EA-IRMS system are described in Fig. S1.These are the same trees and located nearby. Four stems were obtained from each tree, and needles from each tree were mixed and analyzed at Kyoto University (samples for 1999 to 2003) and Hokkaido University (samples after 2004) with Conflo systems (EA 1108 and Delta S, and Flash EA 1112 and Delta V, Thermo Fisher Scientific at Kyoto and Hokkaido Universities respectively). Analytical precisions (standard deviation) of the carbon and nitrogen content measurements were better than 0.3% and 0.1%, respectively, and those for the isotopic compositions $\delta^{13}$C and $\delta^{15}$N were better than 0.2‰. The details of analyses and the average calculations are described in captions of Fig. S1 and S2. In 2015, there were no data on the $\delta^{15}$N and N content. "

Caption of Figure S1: "Temporal variations of raw data in (a) the foliar $\delta^{13}$C and (b) C/N of nine trees in the typical forest during 1999–2019. The four trees LL23, LL24, LLR2, and LLR3 were continuously sampled from 1999 to 2011, and the tree R04 was sampled during 2008–2010, 2014, 2015, and 2017. The four trees S1, S2, S3, and S4 were sampled from 2013. Those sample trees were located nearby. The number of trees sampled for the foliar $\delta^{13}$C (a) every year was: $n$ = 0 in 2012; $n$ = 4 in 1999–2007, 2011, and 2015; $n$ = 5 in 2008–2010 and 2016; $n$ = 6 in 2013, 2018, and 2019; $n$ = 7 in 2017; $n$ = 8 in 2014. The number of trees sampled for the foliar C/N (b)

every year was: $n = 0$ in 2012 and 2015; $n = 4$ in 1999–2007, 2011, 2013, 2014, and 2016–2017; $n = 5$ in 2008–2010; $n = 6$ in 2018 and 2019. From each tree, four stems with current year stems were taken, and leaves on previous and two-year stems were collected in August every year. Needles of each tree were mixed well, kept in a paper bag, and oven-dried at 60 °C in the field. Samples collected before 2004 and after 2004 were brought to Kyoto University and Hokkaido University, respectively, where they were powdered with liquid nitrogen, and oven-dried again. Each sample was then wrapped in a tin capsule and analyzed for carbon and nitrogen contents and for their isotope compositions using the EA-IRMS system. All data obtained in each year were averaged to build a successive temporal variation in the foliar $\delta^{13}C$ and C/N (Fig. 3d and 3f). "

(17) L. 150: can you explain why you chose a pearson correlation, rather than a spearman correlation or crosscorrelation function (which in my experience are more appropriate choices for relatively short timeseries)? Not that I doubt the outcome of your analysis (you present very clear visual and temporal patterns), but the backing of your choices could be stronger.

Since we obtained a simple linear relationship between two parameters, the most common test (Pearson correlation test) was used.

Revision: L. 149-150. "Relationships between two datasets were investigated using a simple linear regression model (function "lm") and a Pearson correlation test ("cor.test"), the most common statistical test based on the method of covariance. "

(18) L. 152: "differences between the two groups" --> which groups are you referring to? there are more than two types of forests mentioned in earlier in the methods. It is also unclear to me why an unpaired test was selected if data from the same years or acquisitions is available for different forests. I am probably misunderstanding what you are describing here, so perhaps that is an indication that better explanaiton is needed.

Two groups mean two different forest types among four (TF, RF-1, RF-2, and DF), but we changed the description about the statistical tests. It is better to use tests comparing 4 forest types at the same time in order to avoid Type I errors. So, we changed statistical method to Kruskal-Wallis test with pairwise Wilcoxon rank sum test. We preferred this test to ANOVA because of a relatively small number of samples in the forest types.

Revisions: L. 152-157 "Differences in NDVI  among four forest types (TF, RF-1, RF-2 and DF) were determined using Kruskal-Wallis test ("kruskal.test") with pairwise Wilcoxon rank sum test ("pairwise.wilcox.test"). ~~using two parametric unpaired two-sample tests, classical Student's and Welch's t tests, and one non-parametric Wilcoxon rank-sum test. The criteria for applying a particular test were the data distribution type (normal or non-normal) and the relation of the data variances to each other (equal or unequal): Student's t-test, both datasets have "normal" distributions and "equal" variances; Welch's t-test, "normal", "unequal"; and Wilcoxon rank-sum test, "non-normal",~~

 The results of the statistical tests are shown in the Supplemental (Table S3 S9)"

We replaced Tables S3, S4 with two new tables, Table S5 was removed:

Table S3. The results of the Kruskal-Wallis test, a non-parametric test to check differences in NDVI among four forest types (TF, RF-1, RF-2, DF), are presented as a significance level $p$-value. The differences in NDVI were significant at $p < 0.05$* (shown in bold font) and insignificant at $p > 0.05$.

Table S3. The results of the Kruskal-Wallis test, a non-parametric test to check differences in NDVI among four forest types (TF, RF-1, RF-2, DF), are presented as a significance level $p$-value. The differences in NDVI were significant at $p < 0.05$* (shown in bold font) and insignificant at $p > 0.05$.

| Date | Kruskal-Wallis test $p$ - value |
|---|---|
| 1999 | 0.261 |
| 2000 | **0.022*** |
| 2001 | **0.002*** |
| 2002 | 0.881 |
| 2003 | 0.258 |
| 2004 | 0.312 |
| 2005 | 0.741 |
| 2006 | 0.063 |
| 2007 | **0.00003*** |
| 2008 | **0.00015*** |
| 2009 | **0.035*** |
| 2010 | **0.004*** |
| 2011 | **0.003*** |
| 2012 | **0.004*** |
| 2013 | **0.00005*** |
| 2014 | **0.013*** |
| 2015 | **0.042*** |
| 2016 | **0.002*** |
| 2017 | **0.003*** |
| 2018 | **0.002*** |
| 2019 | 0.1846 |

Table S4. Comparisons of seasonal maximum NDVI averaged for each forest type among four forest types (TF, RF-1, RF-2, DF) in the years from 1999 to 2019 using pairwise Wilcoxon rank-sum test. The results of the tests are presented as their significance values (*p*-values). Bold font indicates a significant difference flagged as *p* < 0.05.

| Date | Forest types | DF | RF-1 | RF-2 |
|---|---|---|---|---|
| 1999 | RF-1 | 0.31 | | |
| | RF-2 | 1.00 | 1.00 | |
| | TF | 0.31 | 0.31 | 1.00 |
| 2000 | RF-1 | 0.462 | | |
| | RF-2 | 1.000 | 0.275 | |
| | TF | **0.035*** | 0.516 | **0.024*** |
| 2001 | RF-1 | 0.277 | | |
| | RF-2 | 0.800 | 0.207 | |
| | TF | **0.035*** | 0.057 | **0.002*** |
| 2002 | RF-1 | 0.92 | | |
| | RF-2 | 0.92 | 0.92 | |
| | TF | 0.92 | 0.92 | 0.92 |
| 2003 | RF-1 | 0.97 | | |
| | RF-2 | 0.80 | 0.53 | |
| | TF | 0.80 | 0.49 | 0.97 |
| 2004 | RF-1 | 0.77 | | |
| | RF-2 | 0.40 | 0.95 | |
| | TF | 0.79 | 0.57 | 0.40 |
| 2005 | RF-1 | 1.00 | | |
| | RF-2 | 1.00 | 1.00 | |
| | TF | 1.00 | 1.00 | 1.00 |
| 2006 | RF-1 | 0.21 | | |
| | RF-2 | 0.53 | 0.34 | |
| | TF | 0.21 | 0.34 | 0.21 |
| 2007 | RF-1 | 0.123 | | |
| | RF-2 | 0.267 | **0.040*** | |
| | TF | 0.023 | **0.00008*** | **0.001*** |
| 2008 | RF-1 | **0.039*** | | |
| | RF-2 | 0.800 | 0.129 | |
| | TF | **0.023*** | **0.001*** | **0.004*** |
| 2009 | RF-1 | | | |
| | RF-2 | | | |
| | TF | | **0.026*** | |
| 2010 | RF-1 | | | |
| | RF-2 | | 0.750 | |
| | TF | | **0.001*** | 0.667 |
| 2011 | RF-1 | 0.615 | | |
| | RF-2 | 0.640 | 0.661 | |
| | TF | **0.044*** | **0.014*** | **0.007*** |
| 2012 | RF-1 | 0.346 | | |
| | RF-2 | 1.000 | 0.411 | |
| | TF | **0.044*** | **0.026*** | **0.026*** |
| 2013 | RF-1 | 0.031 | | |
| | RF-2 | 0.133 | **0.012*** | |
| | TF | **0.018*** | **0.0007*** | **0.002*** |
| 2014 | RF-1 | 0.62 | | |
| | RF-2 | 0.64 | 0.75 | |
| | TF | 0.05 | 0.05 | 0.05 |
| 2015 | RF-1 | 0.23 | | |
| | RF-2 | 0.23 | 0.34 | |
| | TF | 0.14 | 0.23 | 0.19 |
| 2016 | RF-1 | 0.055 | | |
| | RF-2 | 0.267 | **0.026*** | |
| | TF | **0.026*** | 0.114 | **0.005*** |
| 2017 | RF-1 | 0.073 | | |
| | RF-2 | 0.133 | **0.018*** | |
| | TF | 0.083 | 0.133 | **0.018*** |
| 2018 | RF-1 | **0.040*** | | |
| | RF-2 | 0.533 | **0.040*** | |
| | TF | **0.035*** | 0.088 | **0.008*** |
| 2019 | RF-1 | 0.27 | | |
| | RF-2 | 0.27 | 1.00 | |
| | TF | 0.27 | 1.00 | 1.00 |

(19) L. 163-165: "The seasonal maximum of each year was observed from 25 June to 13 August, except for 1999 (shown in Table S2). The maximum transect NDVI in 1999 was observed on 27 August (0.75 ± 0.02, n = 34) because the Landsat data in 1999 were limited to the latter half of August. " --> landsat scene availability throughout the summer can be highly limited. to what extent is the seasonal maximum an artfeact of data availability (e.g. it would obviously fall in June if no data from July and August are available, even if the true maximum would fall in july or august). Please add an indication or statistical backing (maybe in SI) of how the timing of the seasonal maximum relates to scene availability, because otherwise it cannot be called "year to year variation" and it would be unclear whether the time series you describe in fig. 2a is robust, or merely an artefact of seasonal timing.

[Figure]

Figure above is an example of the time course of NDVI in 2017 summer period. Red triangles and circles are L7 data for transect and 10-km plots, and green triangles and circles are L8 transect and 10-km plots. We divide the growing season to three stages (a) to (c). In June (a), NDVI values quickly increased, and during the late June - the mid-August (or in the beginning of September) (b), NDVI values are relatively stable because vegetation has the maximum biomass. Finally, during the late August-September (c), NDVI values decrease due to the leaf senescence. Although the timings of (b) (start and end) varied depending on the weather and soil moisture conditions, maximum biomass stages (b) continued more than one and half months. We obtained NDVI on this stage as seasonal maximum (the manuscript's supplemental Table S1). The example of NDVI in 2017 (shown in this response letter) is the highest temporal resolution, and other years are lower temporal resolution than that in 2017. But the data in most years had more than 3 data acquisition days in the period of (b). Only one acquisition day during the period of (b) was for 1999 and 2003. For 2003 the observation day was 21 July, and we used this value. For 1999, it was on 27 August. We recognized this NDVI value as seasonal maximum, since this day was in the period of (b). Because of large amount of precipitation in August in 1999, we observed high soil moisture in August 1999 (Figure 3c), and recognized in the period of (b).

We added some sentences between L131 and 132.

Revision:

"The NDVI data of larch forest, that is deciduous, quickly increases in early summer, and during the summer the NDVI is relatively stable (e.g., Huete et al., 2002). This stable NDVI continued more than one and half months (usually from July to mid-August), although the period depends on the weather and soil moisture conditions. Seasonal maximum NDVI in our study was identified during this period. Although the data acquisition days are limited because of low temporal resolution and cloud coverage, more than three days of data acquisition were identified by combination of three satellite images, and seasonal maximum were determined, except for 1999 and 2003. These two years had only one data acquisition day, on 27 August 1999 and 21 July 2003, and both data were recognized as seasonal maximum. "

(20) L. 191 - 192: "To consider the historical variation in the NDVI of typical forests in our study area, the TF NDVI and observed parameters were compared (Fig. 2 and 3)." --> I would strongly urge you to account for landsat scene availability throughout the season, for instance by adding the date within the season as a covariate or interaction. This would give additional information of the association with other parameters may vary across the season and would account for the possibility that the temporal dynamics of ndvi are influenced more by scene availabiltiy than annual dynamics in site conditions.
As we already described in the reply to the comment (19), seasonal maximum during the NDVI stable period was determined in each year, and we believe that the NDVI can be compared with ecosystem parameters.

(21) L. 197: "TF NDVI did not show any correlation with summer temperature" --> you present correlations of NDVI values at different seasonal timings (june / july / august) to overall JJA temperatures. wouldn't it make more sense to compare the ndvi to mean temperatures of degree days up until the moment of ndvi acquisition?
We believe that in the temporal dynamics, the soil moisture and nitrogen availability may be the main environmental factors affecting the NDVI. The summer temperature does affect the NDVI, but in short time periods, e.g., drought events in 2001-2002.

(22) L. 218: the header of the next section accidentally ended up in the figure caption here.
Thank you very much. Next title was mistakenly added.
Revision: "... respectively."

(23) L. 275: "In most years before 2007, the NDVI values in RF and DF were higher than those in TF" --> could this be related to topgraphy; i.e. DF and RF are damaged by floods since they occur in depressions and hence suffer less from drought but more from flooding? the role of terrain is hardly touched upon but potentially very important. It might also be helpful to present

some indication of terrain variability; what is the magnitude of elevation differences between typical DF and Tf sites, for example?

Yes. Before the wet event, soil moisture at RF and DF were higher than TF because of lower elevation. This topographic condition at RF and DF makes lower possibility of drought. We did not observe the altitude in situ, but the difference in elevation between north and south ends by Google Earth Pro was about 5 m. We add the explanation.

Revision:

L. 275 " The difference in elevation between south and north ends of the transect (about 5m in Google Earth) may cause the difference in soil moisture, therefore RF and DF plots showed higher soil moisture and less possibility of drought than TF, before 2007. "

(24) paragraph 4.1: Please discuss whether waterlogging may have influence ndvi directly, independent from tree properties, due to its influence on near infrared reflectance.

Yes. Water shows lower NDVI. We had already described about the possibility of surface water in L281-282. We also add some explanations.

Revision: L. 282 "Water predominantly absorbs NIR radiation and therefore has a low NIR reflectance, resulting in a lower NDVI than vegetation (Holben, 1986)."

(25) L. 312 - 317: I know it is very likely the case, but here you seem to derive causation from the presented correlations. Tone down these causal statements (e.g. "which likely contributed" instead of "which contributed"), or provide more backing for why carbon storage in previous years should be the cause of NDVI dynamics in this period.

We changed the expression.

Revision: L. 312 – 317. "The high level of water availability in the summers of 1999 and 2000 likely contributed to increased carbon storage and, as a result, the high formation of needles in 2000 and 2001. The significant NDVI decrease in 2002 was probably caused by a low level of soil moisture (i.e., dry conditions). The high summer air temperature (Fig. 2c) and the small amount of precipitation (Fig. 2d) in 2001 and 2002 caused droughts in 2002 and 2003. Subsequently, the soil moisture increased due to a large amount of water year precipitation (Fig. 2d), which likely contributed to an increase in NDVI until 2007. "

(26) L. 327 - 328: "The mechanism by which plant δ13 C responds to changes in light and water availability has been well explained in previous studies (e.g., Farquhar et al., 1989). " --> I don't doubt it, but it is very difficult to place your findings on isotope ratios in the appropriate context without some minimum amount of explanation of their meaning and key processes driving isotope fractionation in trees. Please add this (or see comments regarding lines 133-136) at some point so the reader can understand the meaning of the presented work on isotope and c/n ratios to some degree without having to refer to cited work.

The $\delta^{13}C$ value of plant tissue (e.g. leaf) is expressed by the following equation;

$$\delta^{13}C = \delta^{13}C_{atm} - a - (b-a)(C_i/C_a).$$

$\delta^{13}C_{atm}$ is $\delta^{13}C$ of atmospheric $CO_2$, a (4.4‰) and b (27‰) are isotope fractionations of diffusion and enzymatic reaction of photosynthesis (Rubico), and $C_i$ and $C_a$ are inter-cellular and atmospheric $CO_2$ concentrations. When water availability decreases, stomatal conductance increases, which results in the decrease of $CO_2$ incoming to the intercellular, and $C_i$ decreases, resulting in $C_i/C_a$ decrease and $\delta^{13}C$ increase. When water availability increases, $\delta^{13}C$ decrease. For the light condition, when light condition increases, more $CO_2$ is photosynthetically reacted, and Ci decreases, then $\delta^{13}C$ increase. Under low light condition, $C_i$ increases and $\delta^{13}C$ decrease.

We already described what was happening at our site in L329-331. The years 2001 and 2002 were severe drought period (low precipitation and low soil moisture). Under such condition, it is reasonable to consider that stomatal conductance decreased. This is also demonstrated by $\delta^{13}C$ values. Larch tree is deciduous, therefore C photosynthesized in the year makes needles in the next year. Carbon fixed during the drought 2001-2002 makes needles in 2002-2003.

It is not possible to describe the detail explanations above in our main text, but we add the equation and short explanations in the Methods 2.3 after the explanation for comment (16). Besides the following explanation, we also revised the structure of 4.3.1 Water availability, according to the comments (2), and to avoid misunderstanding (see reply of the next comment).

Revision:

"The $\delta^{13}C$ value of plant tissue (e.g. leaf) is expressed by the following equation;

$$\delta^{13}C = \delta^{13}C_{atm} - a - (b - a)(C_i/C_a),$$

$\delta^{13}C_{atm}$ is carbon isotopic composition of atmospheric $CO_2$, a (4.4‰) and b (27‰) are isotope fractionations of diffusion and photosynthetic reaction, and $C_i$ and $C_a$ are inter-cellular and atmospheric $CO_2$ concentrations. In our study site, lower and higher $\delta^{13}C$ values of larch needles were usually arose by wet and drought conditions. "

(27) L. 329: "Under drought stress during 2001–2002, there was a decrease in needle stomatal conductance" --> this is another example of a conclusive statement that does not seem to be backed up by data or a reference. Please check the entire discussion for statements like these and either back them up or tone them down ("has likely decreased stomatal conductance, as suggested by d13C values")

Referee said this statement does not seem to be backed up by data. But for us, "drought -> reducing stomatal conductance ($\delta^{13}C$ increase) -> usually decrease in carbon assimilation" are almost 100% sure. To avoid misunderstanding, we changed the structure of 4.3.1, and added some sentences. We would like to describe that after 2007 "wet condition -> increase in stomatal conductance ($\delta^{13}C$ decrease) -> usually increase in carbon assimilation but actually decrease in carbon assimilation". We observed low NDVI in wet condition, which is probably caused by lower nitrogen availability.

We revised the discussion section "4.3.1. Water availability" as described below:

- L303-325 originally written: describing positive correlation between NDVI and SWE before 2007
- L333-338 originally written: describing negative correlation between NDVI and SWE after 2007
- L 326-331, L338-342 revised and added explanation: describing negative correlation between SWE and $\delta^{13}$C before and after 2007.
- L331-333 originally written: "Comparing the decrease in TF NDVI for drought events, the decrease in TF NDVI for the extreme wet event was not as large (Fig. 2b and 3a), although the extreme wet event caused a significant decrease in the NDVI of RF-1 and RF-2. "
- L342-347 revision: describing that after 2007, usually wet condition makes higher productivity, but low NDVI observed. -> nitrogen availability

Revision: L. 326-347:

"As described in the Sect. 3.3.3, SWE controls forest NDVI because the observation site (northeastern taiga) is established in a continental dry area. We found positive and negative correlations between the NDVI and SWE. Before 2007, the TF NDVI was positively correlated with the June SWE in the current year (Fig. 4b) and positively correlated with the SWE in the previous year June, July, August, and the previous year summer (JJA: June–July–August) (Fig. 4c, 4d, S4c, and S4d, Table S6). This indicates the influence of hydrological conditions in the previous year and early summer of the current year on the leaf productivity of larch trees in the current year.

Larches, as deciduous trees, assimilate carbon through photosynthesis (photoassimilate) during the summer to prepare needles in the next year, and the elongation of needles may be affected by hydrological conditions in the early summer. In the Spasskaya Pad Forest, pulse-labeling experiments with $^{13}CO_2$ showed that stored carbon from the previous year contributed approximately 50 % to formation of new needles in *Larix gmelini* saplings (Kagawa et al., 2006). The high level of water availability in the summers of 1999 and 2000 likely contributed to increased carbon storage and, as a result, the high formation of needles in 2000 and 2001. The significant NDVI decrease in 2002 was probably caused by a low level of soil moisture (i.e., dry conditions). The high summer air temperature (Fig. 2c) and the small amount of precipitation (Fig. 2d) in 2001 and 2002 caused droughts in 2002 and 2003. Subsequently, the soil moisture increased due to a large amount of water year precipitation (Fig. 2d), which likely contributed to an increase in NDVI until 2007.

It is known that the NDVI depends on the previous-year precipitation in arid and semi-arid regions (e.g., Burry et al., 2018; Camberlin et al., 2007). In addition, historical time series of climate indices, based on both precipitation and temperature, were related to one-year lagged NDVI (e.g., Verbyla, 2015; Liu et al., 2017). In boreal interior Alaska, the summer moisture index showed a correlation with maximum summer NDVI not only at a one-year time lag in two 10-km climate station buffers but also at a two-year time lag in many other ones (Verbyla, 2015). Possible reasons for the multi-year NDVI lag could be the long-term negative vegetation responses to drought events, such as a decrease in carbon allocation by plants (e.g., Kannenberg et al., 2019) and plant mortality (e.g., Anderegg et al., 2012). Negative effects of drought events also occurred in our study.

As already mentioned, the positive correlations between the TF NDVI and soil moisture were observed during 1999–2006, however, the correlations were shifted to negative ones during 2008–2019 (Fig. 4b–d

and S4a–e). After 2007, the TF NDVI was negatively correlated with the SWE of all months in the previous (with a one-year time lag) and current years (without a lag) (Table S6). This may indicate that after the extreme wet event, the soil moisture in the previous and current years seemed to negatively affect the current TF NDVI. Therefore, a high level of soil moisture may affect needle production (i.e., carbon assimilation, needle formation, and/or needle elongation).

The mechanism by which plant $\delta^{13}C$ responds to changes in light and water availability has been well explained in previous studies (e.g., Farquhar et al., 1989). In our study site, the $\delta^{13}C$ values of needles usually depend on the water availability (ref Kagawa et al., 2003 JGR, Tei et al., 2019 Ecohydrology). As seen in Figure 5, the significant negative correlation between foliar $\delta^{13}C$ and the previous August SWE was observed during 1999–2007 ($r$ =-0.79, $p < 0.05$; Fig. 5). Interestingly, not only before the wet event, but also for all observation period (1999-2019), negative correlation was found between foliar $\delta^{13}C$ and previous August SWE (Figure 5). These results were different from the correlation between NDVI and SWE, which changed from positive to negative after the wet event.

The negative correlation between foliar $\delta^{13}C$ and SWE in previous August for all observation period (1999-2019) and shift of correlation from positive to negative between TF NDVI and SWE showed the following mechanism. Before the wet event, under drought stress during 2001–2002, there was a decrease in needle stomatal conductance, resulting in a decrease in carbon assimilation. In the subsequent years, 2002–2003, larches produced fewer needles (lower NDVI) with higher $\delta^{13}C$ values from the previously photosynthesized carbon (Fig. 3a and 3d). After the wet event, the correlation between the foliar $\delta^{13}C$ and SWE remained negative, which indicate high stomatal conductance (low foliar $\delta^{13}C$ observed). High stomatal conductance usually contributes the higher potential of a plant to assimilate $CO_2$, store C, and produce needles (high TF NDVI), but after the wet period, larch produced fewer needles (low TF NDVI). Comparing the decrease in TF NDVI for drought events, the decrease in TF NDVI for the extreme wet event was not as large (Fig. 2b and 3a), although the extreme wet event caused a significant decrease in the NDVI of RF-1 and RF-2. The decrease in the TF NDVI in wet years may be due to different factors, in other words nitrogen availability for larches, which can control needle formation, and we will discuss on it in the next chapter."

(28) L. 354 - 346: "Therefore, the decrease in the TF NDVI in wet years may be due to factors other than the carbon assimilation process" --> here you should probably discuss the direct influence of water on near infrared reflectance and ndvi.

As already described in the response to comment (27), we also revised the manuscript. About the effect of surface water, we already described in the response to comment (24).

(29) L. 400 - 401: "However, the TF NDVI and RWI were not significantly correlated after 2007, whereas there was a significant positive correlation before 2007. " --> please consider alternative explanations. For instance, the use of detrending methods in tree ring width series can remove long-term decreases or increases from the time series, and your RWI likley only reflects year-to-year variation in ring width. In this sense, do you think the RWI series reflect any longterm decreases due to for instance waterlogging events and comprimised growth over longer timescales?

As describe by the reviewer, RWI reflects more long-term. There are many interesting things on tree growth. For example, dead trees from waterlogging were affected by not only the waterlogging but also drought several years ago (Tei et al., 2019 Ecohydrolgy). But radial growth of tree is not our aim in our study. Therefore, we cut some sentences (L397-402) and revised L393-402.

Revision: L. 393-402

"The correlation between NDVI and RWI at our observation site was previously reported by Tei et al. (2019b). They used GIMMS-NDVI3g and found its positive correlation with the RWI in the subsequent year during 2004–2014 at the study site. These two parameters, the NDVI and RWI, reflect the carry-over of carbon, which is fixed via needles in the previous year and used in the current year, as experimentally demonstrated by Kagawa et al. (2006).  In previous studies, dendrochronological data showed that tree growth responded to climate with a time lag (e.g., Tei and Sugimoto, 2018). In our study, we could not find a significant correlation between the TF NDVI and RWI at the one-year lag of RWI (Fig. S4g). ~~In our study, soil moisture and nitrogen availability for trees seemed to be the key factors of the environment affecting not only the NDVI, as mentioned above, but also the RWI. However, the TF NDVI and RWI were not significantly correlated after 2007, whereas there was a significant positive correlation before 2007. Thus, the extreme wet event in 2007 could have changed the physiological response of larch trees to the environment in terms of needle and wood production.~~ "

(30) L. 432-434: "To better understand changes in the forest, long-term observation of variations in soil N availability depending on soil moisture and other factors is necessary" --> Perhaps we would also need better understanding and forecasting of precipitation extremes or weather extremes in general?

Yes of course. For the studies of ecosystem change, we need the predictions of climate and weather. But these are totally beyond our aim. So, we did not add the explanation.

(31) L. 435-452: In general, I think the conclusion presents some statements that rely on interpretation quite a lot, and presents a lot of statements that are merely repetition of the results. I do not disagree with your interpretations (I think they are well found), but it should be clear for the reader which statements are interpretations and which are not (e.g. by adding "which we attribute to .."). Also see my main comment; the conclusion does not go beyond the distinct physiological response observed in this ecosystem and does not discuss implications. To be of value to a wide readership, please try to "zoom out" a bit beyond Spasskaya Pad. Maybe mention and discuss the importance of findings such as the long-term alteration of relationships between moisture availability and tree performance, or provide recommendations for future studies.

We will revise the conclusions.

(32) Table 1: The added value of this table relative to the clear patterns in fig 2b, are unclear to me. I also find it unclear why only TF and Rf1 are presented. Due to nestedness (transect plots within years within groups), the p-values should be corrected for pseudoreplication. A visual overview might be stronger here and you could consider replacing or omitting this table.

We removed the Table 1 from the manuscript and revised sentences on L. 235-236 and L. 244-245, which mentioned the Table 1.

Revisions:

L. 235-236: ""

L. 244-245: "During and after 2007, there was no change in the TF NDVI; slightly damaged RF-1 showed a decrease in NDVI to levels similar to those observed during the 2002 drought (Fig. 2b) . "

(33) Figures 4 & 5: "p-values and R2 describe the significance and the degree of variability of the regression  models, respectively" --> degree of variability is probably not the appropriate term here, I assume this is a coefficient of determination?

Yes, this is the coefficient of determination. To avoid misunderstanding, we revised the description.

Revision: Figures 4 and 5 "p-values and $R^2$ describe the significance and  coefficient of determination of the regression model, respectively."

(34) SI tables S4-S5: How reliable are the p values derived for differences among degraded forest and other forest types, if there were only two transect plots with data for degraded forests? I also find it hard to understand why the others use pairwise tests rather than anova/kruskal-wallis tests with post-hoc tests? Throughout the supporting tables S4-S10, you perform very large amounts of t-test and if you want to use these values to support your findings, you should discuss the role of Type I errors.

We changed the statistical test. Please read the response to comment (18).

References, which were used in the response letter.

Anderegg, W. R. L., Berry, J. A., Smith, D. D., Sperry, J. S., Anderegg, L. D. L., and Field, C. B.: The roles of hydraulic and carbon stress in a widespread climate-induced forest die-off, Proceedings of the National Academy of Sciences of the United States of America, 109, 233-237, 10.1073/pnas.1107891109, 2012.

Burry, L. S., Palacio, P. I., Somoza, M., de Mandri, M. E. T., Lindskoug, H. B., Marconetto, M. B., and D'Antoni, H. L.: Dynamics of fire, precipitation, vegetation and NDVI in dry forest environments in NW Argentina. Contributions to environmental archaeology, Journal of Archaeological Science-Reports, 18, 747-757, 10.1016/j.jasrep.2017.05.019, 2018.

Camberlin, P., Martiny, N., Philippon, N., and Richard, Y.: Determinants of the interannual relationships between remote sensed photosynthetic activity and rainfall in tropical Africa, Remote Sensing of Environment, 106, 199-216, 10.1016/j.rse.2006.08.009, 2007.

Evans, R. D.: Physiological mechanisms influencing plant nitrogen isotope composition, Trends in Plant Science, 6, 121-126, 10.1016/s1360-1385(01)01889-1, 2001.

Farquhar, G. D., Ehleringer, J. R., and Hubick, K. T.: Carbon isotope discrimination and photosynthesis, Annual Review of Plant Physiology and Plant Molecular Biology, 40, 503-537, 10.1146/annurev.pp.40.060189.002443, 1989.

Holben, B. N.: CHARACTERISTICS OF MAXIMUM-VALUE COMPOSITE IMAGES FROM TEMPORAL AVHRR DATA, International Journal of Remote Sensing, 7, 1417-1434, 10.1080/01431168608948945, 1986.

Huete, A., Didan, K., Miura, T., Rodriguez, E. P., Gao, X., and Ferreira, L. G.: Overview of the radiometric and biophysical performance of the MODIS vegetation indices, Remote Sensing of Environment, 83, 195-213, 10.1016/s0034-4257(02)00096-2, 2002.

Jin, X. Y., Jin, H. J., Iwahana, G., Marchenko, S. S., Luo, D. L., Li, X. Y., and Liang, S. H.: Impacts of climate-induced permafrost degradation on vegetation: A review, Advances in Climate Change Research, 12, 29-47, 10.1016/j.accre.2020.07.002, 2021.

Kagawa, A., Sugimoto, A., and Maximov, T. C.: Seasonal course of translocation, storage and remobilization of C-13 pulse-labeled photoassimilate in naturally growing Larix gmelinii saplings, New Phytologist, 171, 793-804, 10.1111/j.1469-8137.2006.01780.x, 2006.

Kannenberg, S. A., Novick, K. A., Alexander, M. R., Maxwell, J. T., Moore, D. J. P., Phillips, R. P., and Anderegg, W. R. L.: Linking drought legacy effects across scales: From leaves to tree rings to ecosystems, Global Change Biology, 25, 2978-2992, 10.1111/gcb.14710, 2019.

Liu, J. X., Price, D. T., and Chen, J. A.: Nitrogen controls on ecosystem carbon sequestration: a model implementation and application to Saskatchewan, Canada, Ecological Modelling, 186, 178-195, 10.1016/j.ecolmodel.2005.01.036, 2005.

Liu, S. L., Zhang, Y. Q., Cheng, F. Y., Hou, X. Y., and Zhao, S.: Response of Grassland Degradation to Drought at Different Time-Scales in Qinghai Province: Spatio-Temporal Characteristics, Correlation, and Implications, Remote Sensing, 9, 10.3390/rs9121329, 2017.

Salmon, V. G., Soucy, P., Mauritz, M., Celis, G., Natali, S. M., Mack, M. C., and Schuur, E. A. G.: Nitrogen availability increases in a tundra ecosystem during five years of experimental permafrost thaw, Global Change Biology, 22, 1927-1941, 10.1111/gcb.13204, 2016.

Verbyla, D.: Remote sensing of interannual boreal forest NDVI in relation to climatic conditions in interior Alaska, Environmental Research Letters, 10, 10.1088/1748-9326/10/12/125016, 2015.

Walvoord, M. A. and Kurylyk, B. L.: Hydrologic Impacts of Thawing Permafrost-A Review, Vadose Zone Journal, 15, 10.2136/vzj2016.01.0010, 2016.

---

## Author Response (AR2)

Dear Editor,

We have revised our manuscript according to the comments of the reviewers and Copernicus Editorial Board. This letter contains a point-by-point reply to the comments, including links (line numbers) to the revised manuscript. We referred the line numbers of two revised manuscripts with and without revised parts. Line number of the manuscript with revised parts (marked–up) is shown in parentheses.

Besides, we added our own revisions for the ring width index (RWI), described in the end of the letter, because we added new data. After the update of the RWI dataset, there were no significant changes in the results of our study and their interpretations.

**Reviewer #1.**

We numbered all comments and replied to them.

(1) In the introduction, discussion and particularly the conclusion, the authors mostly discuss earlier findings from Spasskaya Pad, and hardly touch upon potential similarities or dissimilarities with other regions. This makes it very hard for the reader to assess to what extent the findings presented here may hold lessons for the other boreal forests on permafrost. In my view, your results hold important lessons for the potential impacts of increased precipitation variability in northern forests, also beyond Siberian larch forests! Precipitation variability is increasing rapidly in this region (see also https://doi.org/10.1016/j.jhydrol.2021.126865) so it is important to discuss what your findings imply for the future functioning of Siberian larch forests and potentially boreal forests in general. You also demonstrate a clear "legacy effect" that could be related to recent insights regarding duration of the impacts of extremes (see for instance https://onlinelibrary.wiley.com/doi/pdfdirect/10.1111/gcb.16078). You still find divergence in NDVI over ten years after an extreme event. This is a major legacy effect, that has important implications for knowledge on Arctic greening/browning and should be stressed more strongly in the conclusion and abstract!

Response: We added more explanations in the abstract, introduction, discussion section 4.4, and conclusions. In the abstract, the implication of our results was added as described in the reply to the comment (5). In the introduction, we explained the greening and browning trends over high-latitude regions and what factors can control them on L. 38-51 (L. 38-51). Browning was mainly observed in dry regions, including our study site. Besides, introduction was thematically restructured as described in the reply to the comment (8). We explained the long-term effect of the extreme wet event in our study site, which can be potentially observed in other dry regions, in the discussion on 520-529 (L. 475-484) and conclusions on L. 552-554 (L. 504-506).

(2) The described aim of the research is to assess how the local forest has changed over time, but throughout the methods you have decided a priori to split up the data into a pre-2007 and post-2007 segment based on an extreme event. Hence, it seems more appropriate to either first statistically evaluate and demonstrate whether there is a significant trend break. I do not doubt this would be the case if you would try it, but it would provide a back up for your methodological choice. Alternatively (maybe this is easier) you could reframe the research aim to explicitly investigate the effect of this wet event. This would make sense, since the subdivision of forest types within the transect seems to already be based on forest damage and regeneration stadia, and the introduction already extensively discussed observed effects of the 2007 wet period.

Response: Yes, we agree with the comment. Our study is not only the extreme wet event, and we would like to show the historical variation of the larch forest using NDVI. The larch forest at our study and also northeastern Siberian taiga site have been suffered from drought and recently wet event. We would like to show how drought and wet event affect the NDVI. But for the most visible and impacted change in the correlation between NDVI and ecosystem parameter (especially soil moisture) was the wet event. It is not possible to change the statistical analysis at this moment, therefore we reframed the research aim to investigate the effect of wet event as described by the reviewer on L. 103-108 (L. 83-88).

(3) The ecophysiological meaning of the d15N, d13C and C/N ratio data, as well as the methods through which they were derived, are completely lacking. The reader will need more background to understand the presented patterns and the methods are not reproducible here.

Response: We added the explanations. Please read the replies to the comments (15), (16), (26).

(4) I have some concerns about confounding effects of seasonal availability of landsat ndvi data in shaping the temporal dynamics of ndvi and affecting relationships with other site data. In the line comments, I have added some examples and suggestions on how to deal with this. I think with an additional figure or potentially addition of covariates/interactions such issues could be resolved quite well.

Response: When we use the satellite image data, there are many problems such as temporal resolution and combination of different sensors. Please read the replies to the comments (14), (19), and (20). We tried to describe as much as possible.

Answers to the Line comments:

(5) L. 29: Could you reflect briefly on the implications of your results to place them in a wider context? Parts of the Siberian Arctic show record browning in recent decades, as you undoubtedly know better than anyone. Perhaps you could reflect on the potential role of moisture dynamics, drought and waterlogging in this browning trend? (Just a suggestion).

Response: Yes. As you know, boreal forests in northeastern Siberia are experiencing browning, because of not only by temperature-induced drought but also waterlogging and nitrogen dynamics as we showed in our manuscript. We added the sentence to the abstract on L. 31-32 (L. 31-32).

(6) L. 31-32 "occupy a large forest area, approximately 27 % (Fao, 2020)" --> I assume you mean 27% of the world's forest cover? Could you rewrite this to make it clearer what the statistic refers to? Also consider writing "FAO" instead of "Fao" as you also write it in the reference list.

Response: Thank you very much. We changed Fao to FAO. According to FAO, 27% is the percentage of boreal forest in the total forest cover, on L. 35 (L. 35).

(7) L. 39 "and change the ecosystem" --> Could you provide a few concise examples and references?

Response: We removed the sentence after restructuring introduction.

(8) L. 31 - 66: Please consider adding some thematic structuring to the introduction; the introduction seems to give an overview of earlier work that is mostly focused on C-exchange, while the knowledge gap decsribed on L. 65-66 focuses on NDVI and foliar parameters.

Response: We thematically structured the introduction as the following: boreal forests on L. 34-51 (L. 34-51) -> dry Siberian forests on L. 52-67 (L. 52-63) -> not only drought but also extreme wet event affects the forest on L. 68-84 (L. 64-74) -> knowledge gap about NDVI observations on L. 85-91 (L. 75-81) -> aim of the study on L. 102-108 (L. 82-88).

(9) L. 67 - 70: The research aim is described as "assessing how the forest has changed", which seems unnecessarily vague. Could you provide more specific aims or research questions and (optionally) hypotheses? Setting more specific aims may also help provide structure and direction to the introduction paragraph above.

Response: The sentence on L. 103-104 (L. 83-84) described the outline of our aim, and this looks vague. We changed the paragraph as shown on L. 103-108 (L. 82-88).

(10) L. 78: "consists of deciduous species" --> any information which ones? do they occupy a significant share of the canopy compared to dominant larch vegetation?

Response: The deciduous species are larch and birch. To avoid misunderstanding, we changed the description on L. 116-118 (L. 96-98).

(11) L. 80 " and other grasses" --> please remove "other" (as the shrubs mentioned before are not grasses)

Response: Removed on L. 118 (L. 98).

(12) L. 95: "Regenerating forests RF-2 had moderate forest conditions between RF-1 and DF" --> what do you mean by moderate forest conditions?

Response: We described the difference between the RF-2 and RF-1 on L. 130-131 (L. 110-111) and removed the sentence on L. 134 (L. 113-114).

(13) L. 108 - 110: " The transect plots, which consist of pixels not attributed to quality pixels (clear terrain, low-confidence cloud, and low-confidence cirrus) in the quality assessment bit index band according to Landsat Surface Reflectance product guides, were excluded from the analysis. -> due to the structure of this sentence it reads to me as though all transect plots ndvi values were excluded from analysis, but as the text continuous you describe how it was used in further analysis, so I assume you only removed pixels (or transect plots?) that were flagged in the QA product? Perhaps you could rephrase this more clearly (e.g. that "pixels flagged in the quality assessment bands were omitted from analysis"? or that "transect plots that contained pixels flagged in the quality assessment bands were omitted from analysis"?).

Response: We rephrased the sentence on 148-150 (L. 128-130).

(14) L. 120: can you provide an assessment of fit among the different sensors, e.g. on days for which multiple products are available? how accurate is the estimate for the one sensor based on another sensor compared to the actual value? Roy et al 2016 recommend to use a locally parameterized regression, although it would be understandable if insufficient overlap in acquisitions among different sensors prevents establishment of specific regression parameters for your site.

Response: We understand that local parameterization is important, because it is not possible to combine different sensors perfectly. However, unfortunately, we cannot show the sufficient data of assessments for publication. In our study, three Landsat images (Landsat 5 TM (L5), Landsat 7 ETM+ (L7), and Landsat 8 OLI (L8)) were available. L7 had the longest observation period, but actually data quality was not so good, compared to L5 and L8 (after the scan-line corrector failure of L7 in 2003). After the selection of image data and conversion by Roy et al. (2016) and Ju and Masek (2016) as described in Methods 2.2, and we again selected the NDVI data for comparisons between L7 and L5 for the period 1999-2011, and L8 and L7 for 2013-2019, by the following conditions.

- For transect plots, all 34 transect plots were observed. For 10-km plot, more than 96% of pixels in L5 and L8, and more than 75% of pixels in L7 were observed.
- There was one day difference in the acquisition dates between L5 and L7 and between L7 and L8, and NDVI signals were close.
- If the average value for the short period in summer (NDVI shows usually small change in July to beginning of August) was calculated, we used the average value.

Eleven data (including transect and 10-km plot) for comparison between L7 and L5 and twelve data (including transect and 10-km plot) between L8 and L7 were identified. The results were close to the 1:1 line (see the figure below).

There are many problems on statistical procedure if we show these assessments in our paper. But
we believe that the conversions by Roy et al. (2016) and Ju and Masek (2016) can be used
realistically.

We put some sentences to Methods 2.2 L. 160-163 (L. 140-143) and Discussion 4.1 L. 365-
367 (L. 337-339).

[Figure]

(15) L. 133-136: this paragraph lacks context of the ecological or physiological meaning of
isotope ratios and C/N ratio. More explanation and literature is needed for the non-expert reader
to assess what the d15N, d13C, C/N ratios and ring widths actually mean and what questions you
are answering by including these data (alternatively, you could also already explain how the
different types of datasets relate to the research aims in the final introduction paragraph)

Response: We revised and added some sentences for explanation and literature on L. 184-
199, 209-211 (L. 164-179, 189-191).

(16) L. 133-136: There seems to be no explanation of how the d15N, d13C and C/N ratios
were derived, Add methodology (which tissues were sampled, how many grams, how were they
analyzed, on which instrument, against which isotope standards at what precision?). If the data come from an existing dataset or study, please cite it so the reader can understand how the values
were derived.

Response: We put the outline of methodology in Methods 2.3 L. 200-207 (L. 180-187) and
the methodology details in the caption of Figure S1.

(17) L. 150: can you explain why you chose a pearson correlation, rather than a spearman
correlation or crosscorrelation function (which in my experience are more appropriate choices for
relatively short timeseries)? Not that I doubt the outcome of your analysis (you present very clear
visual and temporal patterns), but the backing of your choices could be stronger.

Response: Since we obtained a simple linear relationship between two parameters, the most
common test (Pearson correlation test) was used. We added the short description on L. 221-222
(L. 201-202).

(18) L. 152: "differences between the two groups" --> which groups are you referring to?
there are more than two types of forests mentioned in earlier in the methods. It is also unclear to
me why an unpaired test was selected if data from the same years or acquisitions is available for
different forests. I am probably misunderstanding what you are describing here, so perhaps that is
an indication that better explanaiton is needed.

Response: Two groups mean two different forest types among four (TF, RF-1, RF-2, and
DF), but we changed the description about the statistical tests on L. 223-228 (L. 203-205). It is
better to use tests comparing 4 forest types at the same time in order to avoid Type I errors. So,
we changed statistical method to Kruskal-Wallis test with pairwise Wilcoxon rank sum test. We
preferred this test to ANOVA because of a relatively small number of samples in the forest types.
In the manuscript supplemental, we replaced the old Tables S3, S4 with two new tables, and
the old Table S5 was removed. Consequently, we changed the table number on L.229 (L. 205),
296 (274), 311 (288), 326 (298), 327 (299), 330 (302), 344 (316), 383 (355).

(19) L. 163-165: "The seasonal maximum of each year was observed from 25 June to 13
August, except for 1999 (shown in Table S2). The maximum transect NDVI in 1999 was observed
on 27 August ($0.75 \pm 0.02$, n = 34) because the Landsat data in 1999 were limited to the latter half of August. " --> landsat scene availability throughout the summer can be highly limited. to what extent is the seasonal maximum an artfeact of data availability (e.g. it would obviously fall in June if no data from July and August are available, even if the true maximum would fall in july or august). Please add an indication or statistical backing (maybe in SI) of how the timing of the seasonal maximum relates to scene availability, because otherwise it cannot be called "year to year variation" and it would be unclear whether the time series you describe in fig. 2a is robust, or merely an artefact of seasonal timing.

[Figure]

Response: The figure above is an example of the time course of NDVI in 2017 summer period. Red triangles and circles are L7 data for transect and 10-km plots, and green triangles and circles are L8 transect and 10-km plots. We divide the growing season to three stages (a) to (c). In

June (a), NDVI values quickly increased, and during the late June - the mid-August (or in the beginning of September) (b), NDVI values are relatively stable because vegetation has the maximum biomass. Finally, during the late August-September (c), NDVI values decrease due to the leaf senescence. Although the timings of (b) (start and end) varied depending on the weather and soil moisture conditions, maximum biomass stages (b) continued more than one and half months. We obtained NDVI on this stage as seasonal maximum (the manuscript's supplemental

Table S1). The example of NDVI in 2017 (shown in this response letter) is the highest temporal resolution, and other years are lower temporal resolution than that in 2017. But the data in most years had more than 3 data acquisition days in the period of (b).  Only one acquisition day during the period of (b) was for 1999 and 2003. For 2003 the observation day was 21 July, and we used this value. For 1999, it was on 27 August. We recognized this NDVI value as seasonal maximum, since this day was in the period of (b). Because of large amount of precipitation in August in 1999, we observed high soil moisture in August 1999 (Figure 3c), and recognized in the period of (b).

We added some sentences on L. 176-182 (L. 156-162).

(20) L. 191 - 192: "To consider the historical variation in the NDVI of typical forests in our study area, the TF NDVI and observed parameters were compared (Fig. 2 and 3)." --> I would strongly urge you to account for landsat scene availability throughout the season, for instance by adding the date within the season as a covariate or interaction. This would give additional information of the association with other parameters may vary across the season and would account for the possibility that the temporal dynamics of ndvi are influenced more by scene availabiltiy than annual dynamics in site conditions.

Response: As we already described in the reply to the comment (19), seasonal maximum during the NDVI stable period was determined in each year, and we believe that the NDVI can be compared with ecosystem parameters.

(21) L. 197: "TF NDVI did not show any correlation with summer temperature" --> you present correlations of NDVI values at different seasonal timings (june / july / august) to overall JJA temperatures. wouldn't it make more sense to compare the ndvi to mean temperatures of degree days up until the moment of ndvi acquisition?

Response: We believe that in the temporal dynamics, the soil moisture and nitrogen availability may be the main environmental factors affecting the NDVI. The summer temperature does affect the NDVI, but in short time periods, e.g., drought events in 2001-2002.

(22) L. 218: the header of the next section accidentally ended up in the figure caption here.

Response: Thank you very much. The next title was mistakenly added. It was removed on L. 287 (L. 265).

(23) L. 275: "In most years before 2007, the NDVI values in RF and DF were higher than those in TF" --> could this be related to topgraphy; i.e. DF and RF are damaged by floods since they occur in depressions and hence suffer less from drought but more from flooding? the role of terrain is hardly touched upon but potentially very important. It might also be helpful to present
some indication of terrain variability; what is the magnitude of elevation differences between
typical DF and Tf sites, for example?
Response: Yes. Before the wet event, soil moisture at RF and DF were higher than TF
because of lower elevation. This topographic condition at RF and DF makes lower possibility of
drought. We did not observe the altitude in situ, but the difference in elevation between north and
south ends by Google Earth Pro was about 5 m. We added the explanation on L. 345-348 (L. 317-
320).

(24) paragraph 4.1: Please discuss whether waterlogging may have influence ndvi directly,
independent from tree properties, due to its influence on near infrared reflectance.
Response: Yes. Water shows lower NDVI. We had already described about the possibility
of surface water on L. 353-354 (L. 325-326). We also added some explanation on L. 354-355 (L.
326-327).

(25) L. 312 - 317: I know it is very likely the case, but here you seem to derive causation
from the presented correlations. Tone down these causal statements (e.g. "which likely
contributed" instead of "which contributed"), or provide more backing for why carbon storage in
previous years should be the cause of NDVI dynamics in this period.
Response: We changed the expressions, e.g. on L. 388-393 (L. 360-365).

(26) L. 327 - 328: "The mechanism by which plant $\delta13$ C responds to changes in light and
water availability has been well explained in previous studies (e.g., Farquhar et al., 1989). " --> I
don't doubt it, but it is very difficult to place your findings on isotope ratios in the appropriate
context without some minimum amount of explanation of their meaning and key processes driving
isotope fractionation in trees. Please add this (or see comments regarding lines 133-136) at some
point so the reader can understand the meaning of the presented work on isotope and c/n ratios to
some degree without having to refer to cited work.
Response: The $\delta^{13}C$ value of plant tissue (e.g. leaf) is expressed by the following equation:
$\delta^{13}C = \delta^{13}C_{atm} - a - (b-a)(C_i/C_a)$.

$\delta^{13}C_{atm}$ is $\delta^{13}C$ of atmospheric $CO_2$, a (4.4‰) and b (27‰) are isotope fractionations of diffusion and enzymatic reaction of photosynthesis (Rubisco), and $C_i$ and $C_a$ are inter-cellular and atmospheric $CO_2$ concentrations. When water availability decreases, stomatal conductance increases, which results in the decrease of $CO_2$ incoming to the intercellular, and $C_i$ decreases, resulting in $C_i/C_a$ decrease and $\delta^{13}C$ increase. When water availability increases, $\delta^{13}C$ decrease.

For the light condition, when light condition increases, more $CO_2$ is photosynthetically reacted, and Ci decreases, then $\delta^{13}C$ increase. Under low light condition, $C_i$ increases and $\delta^{13}C$ decrease.

We already described what was happening at our site during the drought. The years 2001

and 2002 were severe drought period (low precipitation and low soil moisture). Under such condition, it is reasonable to consider that stomatal conductance decreased. This is also demonstrated by $\delta^{13}C$ values. Larch tree is deciduous, therefore C photosynthesized in the year makes needles in the next year. Carbon fixed during the drought 2001-2002 makes needles in

2002-2003.

It is not possible to describe the detailed explanations above in our main text, but we add the equation and short explanations in the Methods 2.3 after the explanation for comment (16). Besides the following explanation, we also revised the structure of 4.3.1 Water availability, according to the comment (2), and to avoid misunderstanding (see the reply to the next comment).

(27) L. 329: "Under drought stress during 2001–2002, there was a decrease in needle stomatal conductance" --> this is another example of a conclusive statement that does not seem to be backed up by data or a reference. Please check the entire discussion for statements like these and either back them up or tone them down ("has likely decreased stomatal conductance, as suggested by d13C values")

Response: Referee said this statement does not seem to be backed up by data. But for us,

"drought -> reducing stomatal conductance ($\delta^{13}C$ increase) -> usually decrease in carbon assimilation" are almost 100% sure. We would like to describe that after 2007 "wet condition ->

increase in stomatal conductance ($\delta^{13}C$ decrease) -> usually increase in carbon assimilation but actually decrease in carbon assimilation". We observed low NDVI in wet condition, which is probably caused by lower nitrogen availability. To avoid misunderstanding, we changed the structure of 4.3.1, and added some sentences on 402-434 (L. 374-394).

(28) L. 354 - 346: "Therefore, the decrease in the TF NDVI in wet years may be due to factors other than the carbon assimilation process" --> here you should probably discuss the direct influence of water on near infrared reflectance and ndvi.

Response: As already described in the response to comment (27), we also revised the manuscript. About the effect of surface water, we already described in the response to comment (24).

(29) L. 400 - 401: "However, the TF NDVI and RWI were not significantly correlated after 2007, whereas there was a significant positive correlation before 2007. " --> please consider alternative explanations. For instance, the use of detrending methods in tree ring width series can remove long-term decreases or increases from the time series, and your RWI likley only reflects year-to-year variation in ring width. In this sense, do you think the RWI series reflect any long-term decreases due to for instance waterlogging events and comprimised growth over longer timescales?

Response: As describe by the reviewer, RWI reflects more long-term. There are many interesting things on tree growth. For example, dead trees from waterlogging were affected by not only the waterlogging but also drought several years ago (Tei et al., 2019 Ecohydrology). But radial growth of tree is not our aim in our study. Therefore, we cut some sentences on L. 487-491 and revised L. 486-487 (L. 445-446).

(30) L. 432-434: "To better understand changes in the forest, long-term observation of variations in soil N availability depending on soil moisture and other factors is necessary" --> Perhaps we would also need better understanding and forecasting of precipitation extremes or weather extremes in general?

Response: Yes, of course. For the studies of ecosystem change, we need the predictions of climate and weather. But these are totally beyond our aim. So, we did not add the explanation.

(31) L. 435-452: In general, I think the conclusion presents some statements that rely on interpretation quite a lot, and presents a lot of statements that are merely repetition of the results. I do not disagree with your interpretations (I think they are well found), but it should be clear for the reader which statements are interpretations and which are not (e.g. by adding "which we attribute to .."). Also see my main comment; the conclusion does not go beyond the distinct physiological response observed in this ecosystem and does not discuss implications. To be of value to a wide readership, please try to "zoom out" a bit beyond Spasskaya Pad. Maybe mention and discuss the importance of findings such as the long-term alteration of relationships between moisture availability and tree performance, or provide recommendations for future studies.

Response: We revised the conclusions on L. 534-564 (L. 489-506). The obtained results and their interpretations were explained together, and the expressions of the interpretations were changed. The implication of the results was also added.

(32) Table 1: The added value of this table relative to the clear patterns in fig 2b, are unclear to me. I also find it unclear why only TF and Rf1 are presented. Due to nestedness (transect plots within years within groups), the p-values should be corrected for pseudoreplication. A visual overview might be stronger here and you could consider replacing or omitting this table.

Response: We removed the Table 1 from the manuscript and revised sentences on L. 304-305 and L. 400-401 (L. 372-372), which mentioned the Table 1.

(33) Figures 4 & 5: "p-values and R2 describe the significance and the degree of variability of the regression models, respectively" --> degree of variability is probably not the appropriate term here, I assume this is a coefficient of determination?

Response: Yes, this is the coefficient of determination. To avoid misunderstanding, we revised the description on L. 286-287 (L. 264-265) and L. 340-341 (L. 312-313), the caption of Figure S4.

(34) SI tables S4-S5: How reliable are the p values derived for differences among degraded forest and other forest types, if there were only two transect plots with data for degraded forests? I also find it hard to understand why the others use pairwise tests rather than anova/kruskal-wallis tests with post-hoc tests? Throughout the supporting tables S4-S10, you perform very large amounts of t-test and if you want to use these values to support your findings, you should discuss the role of Type I errors.

Response: We changed the statistical test. Please read the response to the comment (18).

Reviewer #2.

In "Historical variation in normalized difference vegetation index compared with soil moisture at a taiga forest ecosystem in northeastern Siberia" the authors investigated the variation in NDVI among forest conditions (typical mature, TF; regenerating-1, RF-1; regenerating-2, RF-

2; and damaged forests, DF) and field-observed parameters (from 1998 to 2019) such as RWI, soil moisture, changes of larch needles ($\delta$13C, $\delta$15N, C/N), air temperature, and precipitation. The authors determined that prior to the 2007 extreme wet event, wet areas like DF and RF had higher

NDVI values than dry TF sites due to greater water availability. However, following 2007, the TF

had a greater NDVI than the DF and RF, although being visibly unaffected by the wet event.

Studying historical variations in NDVI compared with soil moisture at a taiga forest ecosystem in north-eastern Siberia is important for several reasons. Firstly, NDVI data can provide valuable information about temporal and spatial changes in vegetation distribution, productivity, and dynamics, which allows for the monitoring of habitat degradation and fragmentation. Secondly, the comparison of historical variations in NDVI with soil moisture can provide insights into the impact of extreme weather events on vegetation, such as the extreme wet event in 2007, which resulted in high tree mortality and a decrease in NDVI at affected sites. Understanding the ecological effects of climatic disasters such as drought or fire can be assessed using NDVI data, making it a valuable tool for monitoring changes in vegetation due to climate change. Overall, studying historical variations in NDVI and soil moisture in a taiga forest ecosystem can provide valuable insights into the impact of extreme weather events on vegetation and the effects of climate change on vegetation dynamics. Therefore, this paper has the potential to make an important contribution to the body of knowledge concerning the impacts of global change on sensitive and complex permafrost ecosystems.

It is my opinion that the authors used sound methods to address the study aims and presented the research findings clearly and concisely and they used appropriate figures to illustrate the NDVI

values of the forest types and the trends in the transect and 10-km plot, which could be useful for researchers and policymakers. However, I agree with referee 1 about their main points raised as well as the minor comments provided. To avoid repetition and in the interest of brevity, I will not be going over them again in this review, but I strongly advise the authors to make the corrections
already suggested. Instead, I will just add a few points concerning the discussion section that I would like to see addressed before publication. When the authors revise these issues, I recommend the study for publication in Biogeosciences.

In the discussion, the authors considered the probable reasons for the differences in NDVI values among the forest types, such as the change in vegetation and the presence of surface water and saturated soil. However, the section could benefit from a more critical evaluation of the results and their implications. For example, the article does not address the limitations of using NDVI as a proxy for vegetation health and productivity, which could impact the accuracy of the results. NDVI measures the amount of chlorophyll in the uppermost layers of vegetation. This means that it may not accurately represent the health and productivity of plants with lower canopies or those that are hidden from view. The limitations of using NDVI as a proxy for vegetation health and productivity may be particularly relevant in taiga/permafrost ecosystems due to their complex vegetation structure and sensitivity to environmental changes.

Additionally, the article does not explore the broader ecological implications of these findings, such as how changes in vegetation health and productivity may impact ecosystem services or the ability of forests to sequester carbon. Finally, while the article notes the potential for using the observational data for analyses of ecosystem changes at the plot and regional scales, it does not explicitly state what these analyses might entail or why they would be valuable. A more explicit discussion of the practical applications of the research could make the findings more accessible to a wider audience.

Response: We added the explanations for the limitations of using NDVI on L. 85-91, L. 365-367 (L. 337-339), L. 520-523 (L. 475-478). The NDVI was shown to be affected not only by the overstory vegetation but also the understory vegetation. But our study was mainly focused on the typical larch forest, which was not visibly damaged by the extreme wet event. The typical forest presumably showed higher contribution of the overstory (larches) to the NDVI than damaged forests. The NDVI and larch needle C/N showed a significant correlation, so it is likely that NDVI showed the overstory conditions.

We also added some descriptions about explicit discussion of the practical applications in the discussion 4.4 L. 520-532 (L. 475-487). This phenomenon observed at our study site might happen in other dry regions. The implication of our results was also shown in the abstract L. 31-
32 (L. 31-32) and conclusions on L. 552-554 (L. 504-506).

Copernicus editorial.

(1) I noticed that your Figure 1(a) contains a map.
For the next revision, I kindly ask you to clarify whether you have created the maps or were
they created by a map provider?
If the maps were not created by you, please provide in your revised file that the copyright is
denoted in the figure itself. If this is not possible, please provide it in the caption.
Response: We added descriptions about the providers of the maps used in Figures 1 (a) and 1 (b)
in the caption of Figure 1 on L. 137-138 (L.117-118) and on Figure 1a itself.

(2) Before file upload, please consider submitting data sets, model code, or video supplements
to reliable repositories, receive DOIs, and cite these assets in your manuscript including entries in
the reference list.
Response: We added information about the datasets used in our study in the "Data availability"
and "References" sections.

Authors' revisions.

We updated the ring width index (RWI) dataset after adding a greater number of larch tree paired
increment core samples. As a result, the results of our study, namely the descriptions of Pearson
correlation (r, p-value) and linear regression models ($R^2$, p-value) between the TF NDVI and RWI,
after the update were not significantly different from those before the update, so there were no
changes in their interpretations. The revisions were made in the results section 3.3.2 on L. 289-297
(L. 267-275), the discussion section 4.3.3 on L. 473 (L. 433), Figures 3b and 4a, and in the
supplemental Figure S4g, Table S2 and S5-S7.